# IN-CONTEXT ALGEBRA

**Eric Todd**[1]* **Jannik Brinkmann**[1,2] **Rohit Gandikota**[1] **David Bau**[1]
[1]Northeastern University [2]TU Clausthal

## ABSTRACT

We investigate the mechanisms that arise when transformers are trained to solve arithmetic on sequences where tokens are *variables* whose meaning is determined only through their interactions in-context. While prior work has studied transformers in settings where the answer relies on fixed parametric or geometric information encoded in token embeddings, we devise a new in-context reasoning task where the assignment of tokens to specific algebraic elements varies from one sequence to another. Despite this challenging setup, transformers achieve near-perfect accuracy on the task and even generalize to unseen groups. We develop targeted data distributions to create causal tests of a set of hypothesized mechanisms, and we isolate three mechanisms models consistently learn: commutative copying where a dedicated head copies answers, identity element recognition that distinguishes identity-containing facts, and closure-based cancellation that tracks group membership to constrain valid answers. Our findings show that the kinds of reasoning strategies learned by transformers are dependent on the task structure and that models can develop symbolic reasoning mechanisms when trained to reason in-context about variables whose meanings are not fixed.

## 1 INTRODUCTION

The hallmark of abstract reasoning is the ability to work with words and symbols whose meaning is unknown ahead of time (Newell, 1980; Lampinen et al., 2024). Yet much of the performance of language models (LMs) can be attributed to the power of the token embedding, for example pre-encoding the attribute *female* in the embedding for "Queen," (Mikolov et al., 2013) or pre-encoding *divisible-by-two* within the embedding of the token "108" (Zhou et al., 2024; Hu et al., 2025; Kantamneni & Tegmark, 2025; Nikankin et al., 2025). What mechanisms can a transformer language model employ if it is unable to pre-encode solutions in the embeddings of the words?

In this work, we devise a simple in-context learning setting where tokens serve as pure variables, acquiring meaning only through their interactions within each sequence. This allows us to ask: What computational strategies do transformers develop when deprived of meaningful embeddings?

We adopt a familiar arithmetic problem setting, training small transformers to predict answers to arithmetic problems sampled from finite groups. What makes our setting unique is that each token is a variable that can represent any algebraic element: the meaning of each token is only fixed within a single sequence. Complementary to previous studies of emergent arithmetic reasoning (Power et al., 2022; Zhang et al., 2022; Nanda et al., 2023; Zhong et al., 2023), solving problems in this setting forces models to infer structure solely from observations of contextual relationships (Piantadosi & Hill, 2022) rather than relying on fixed information encoded within token embeddings.

Surprisingly, we find that models trained on this task develop fundamentally different reasoning strategies than those previously observed when LMs solve arithmetic in fixed-token settings. Rather than learning geometric representations of a Fourier basis, we find that models acquire symbolic reasoning mechanisms based on sparse relational patterns (Zucchet et al., 2025). We identify three primary algorithmic strategies models employ beyond verbatim copying: commutative copying, identity element recognition, and closure-based cancellation. These findings suggest that the kinds of reasoning strategies learned by transformers are dependent on the task structure, with symbolic, context-based strategies emerging when tokens carry no consistent meaning across contexts.

---

*Correspondence to todd.er@northeastern.edu. Open-source code and data available at algebra.baulab.info.

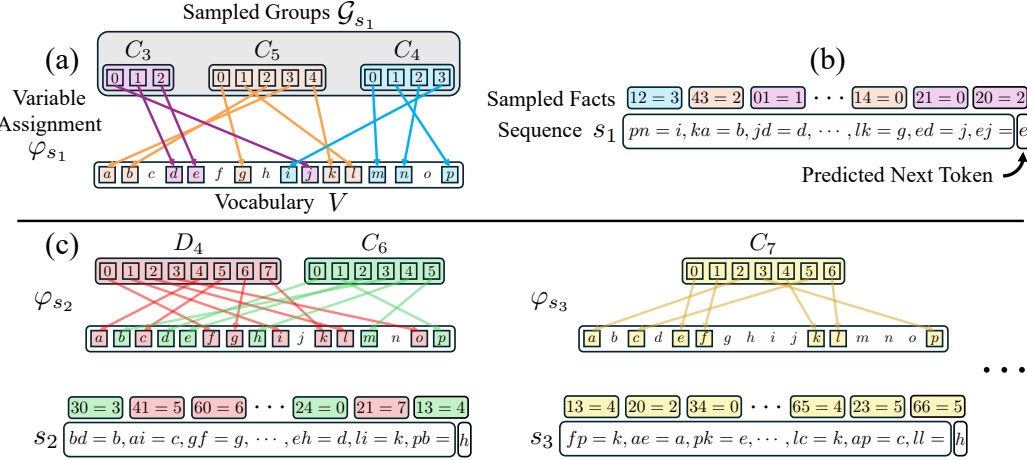

Figure 1: An overview of the data generation process. (a) **Variable Assignment:** We sample a set of finite groups and assign the elements of each group a non-overlapping set of vocabulary symbols. (b) **Sequence Generation:** Sampled facts are converted into variable statements via the latent mapping $\varphi_s$ and concatenated together to form a sequence. (c) **Sample Diversity:** Every sequence is constructed by sampling a new set of groups, defining a new latent mapping, and sampling a new string of facts. The vocabulary symbols are assigned specific meanings within individual sequences, but can take on very different meaning across sequences.

## 2 TASK DESCRIPTION

In this section, we describe our in-context learning task. At a high level, our task involves simulating a mixture of finite groups.[1] Each task sequence presents several examples of products between elements in a group and the model is trained on the ordinary next-token language modeling objective, with the goal that it will learn to predict the outcome of unseen group products (Figure 1).

More formally, we have a set of $m$ groups $\mathcal{G} = \{G_1, G_2, \ldots, G_m\}$ that the model is trained to simulate. Recall that for any finite group $G$, the product of two elements $x, y \in G$ is written as $z = x \cdot y \in G$. We call each such product "$x \cdot y = z$" a *fact*. Training data consists of sequences of $k$ facts written using a vocabulary of variable tokens $v_i \in V$ whose meaning may vary between sequences. In practice, the vocabulary is small, with $N = |V| < \sum_i |G_i|$. A typical sequence $s$ takes the form shown in Equation 1, where individual facts consist of four tokens.

$$s = \text{``}v_{x1}v_{y1} = v_{z1}, \ v_{x2}v_{y2} = v_{z2}, \ \cdots, \ v_{xk}v_{yk} = v_{zk}\text{''} \tag{1}$$

We describe the positions of a fact with the following terminology: The first element $v_{xi}$, occupies the "left-slot"; $v_{yi}$ is the "right-slot"; = is the "predictive token"; and $v_{zi}$ is in the "answer-slot".

To generate a training sequence $s$, we first sample[2] a set of groups $\mathcal{G}_s$ from $\mathcal{G}$ whose total number of elements is less than or equal to the number of variable tokens $N$. We define the set of all sampled group elements to be $H_s = \bigcup \mathcal{G}_s$, where $|H_s| \leq N$. We then construct a one-to-one latent mapping $\varphi_s : H_s \to V$ that randomly assigns all elements of $H_s$ to distinct tokens in $V$. We ensure that each group in $\mathcal{G}_s$ is assigned a non-overlapping set of variables so that the meaning of each variable within a given sequence is determined by the underlying group structure (Figure 1a).

Given this latent mapping, we then assemble $s$ by sampling facts from the groups in $\mathcal{G}_s$, converting them to variable statements via $\varphi_s$, and concatenating them together (Figure 1b). The statement "$\varphi_s(x)\varphi_s(y) = \varphi_s(z)$" only appears in $s$ when there is a corresponding valid fact "$x \cdot y = z$" among the sampled groups in $\mathcal{G}_s$. Importantly, while the mapping $\varphi_s$ is fixed within a sequence, it varies between sequences, ensuring that vocabulary tokens $v_i \in V$ act as variables without fixed global meaning (Figure 1c).

---

[1]While not imperative for understanding our task setup, we provide a brief review of relevant topics from group theory in Appendix A.

[2]For more details about how we sample groups to construct $\mathcal{G}_s$, see Appendix C.2.

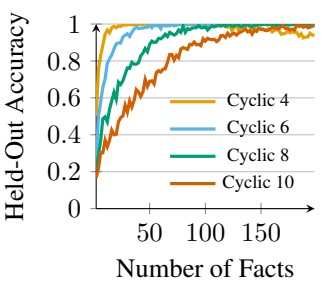 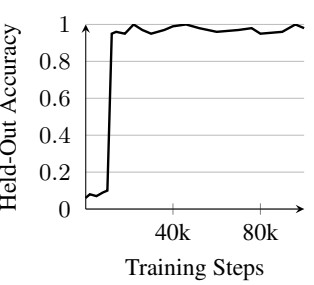 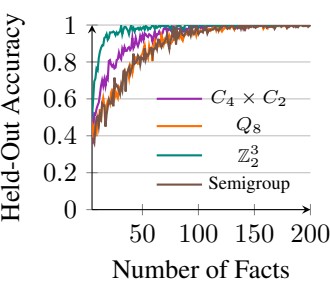

(a) **Performance increases with context length.** Accuracy increases monotonically with the number of in-context facts. Groups of higher order require more in-context facts to achieve near-perfect performance.

(b) **Phase transition on non-copyable facts.** Accuracy on queries where copying is impossible remains low early in training but then rises abruptly, indicating that the model learns to generalize beyond simple copying strategies.

(c) **Generalization across algebraic structure.** The model generalizes to unseen groups of order 8 and also achieves non-trivial hold-out accuracy on semigroups.

Figure 2: In-context algebra performance.

## 3 CAN TRANSFORMERS LEARN IN-CONTEXT ALGEBRA?

We train transformer models on the in-context algebra task (§2) and evaluate both their in-distribution performance and their ability to generalize across contexts. We report results for one representative model throughout, but observe qualitatively similar patterns across multiple training runs (see Figure 7 in Appendix C). Our main model is a 4-layer autoregressive transformer with 8 attention heads per layer and hidden size 1024, trained with next-token prediction on sequences of $k=200$ algebra facts ($\sim 1000$ tokens). The training distribution $\mathcal{G} = \{C_3, \ldots, C_{10}, D_3, D_4, D_5\}$ includes cyclic ($C_i$) and dihedral groups ($D_i$) of up to order 10, with sequences written using $N=16$ variable tokens plus the special tokens '=' and ','. Because group-to-variable assignments are randomized per sequence, tokens act as placeholders whose meaning must be inferred from context.

**Performance increases with context length.** Accuracy increases monotonically with the number of in-context facts $k$, but the rate of improvement depends on the group order (Fig. 2a). Smaller groups (e.g., $C_4$, $C_6$) reach high accuracy with only a few facts, whereas larger groups (e.g., $C_8$, $C_{10}$) require substantially more context to achieve similar performance (see Fig. 10 in Appendix C.3).

**Phase transition on non-copyable queries.** At large $k$, many queries are trivially solvable by copying a previously seen fact (about $90\%$ of queries are copyable at $k=200$ versus $45\%$ at $k=50$). To address this, we evaluate the model with held-out data where the final fact "$xy=$" and its commutative pair "$yx=$" never appear elsewhere in the sequence. In this setting, the model still achieves near perfect accuracy, and we observe an abrupt improvement during training (a phase transition) on non-copyable queries (Fig. 2b), suggesting the emergence of strategies beyond verbatim retrieval.

**Generalization across algebraic structure.** The model also transfers to unseen groups: on the complete set of order-8 groups (for groups excluded from training), the model also achieves near-perfect performance (Fig. 2c). Interestingly, hold-out performance is good for non-group structures such as semigroups, but is worse for quasigroups and collapses on magmas (Fig. 11b in Appendix C.3). The model still achieves non-trivial accuracy for quasigroups, particularly on cancellation data (Fig. 12 in Appendix C.3), though generalization remains consistently stronger for groups than non-groups.

## 4 HYPOTHESIZING MODEL MECHANISMS

When analyzing a random in-context algebra sequence, it is possible that multiple algorithms could theoretically produce correct predictions. That can make it challenging to identify which mechanisms the model actually implements. Consider the sequences shown in Equation 2 and Equation 3 that differ only by which fact is bolded. The model could correctly predict "$dp=\boldsymbol{p}$" by either copying the answer from the duplicate fact appearing earlier (i.e., $dp=p$) or by recognizing $d$ is an identity

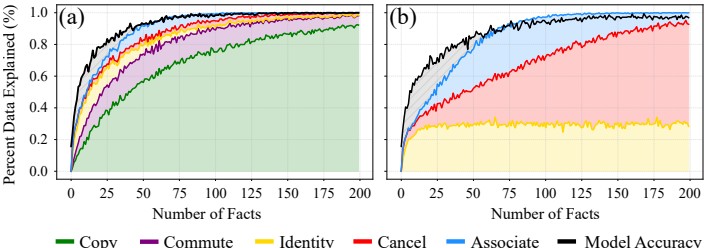

Figure 3: Algorithmic coverage: **(a)** the percentage of training data that can be solved by each mechanism: copying (green), commutative copying (purple), identity recognition (yellow), closure-based elimination (red), associativity (blue), compared to the empirical model performance (black). The gray shaded region represents unexplained performance. **(b)** Coverage of sequences where neither form of copying is possible. Identity recognition solves 28.7% of the problems (yellow), closure-based cancellation can solve an additional 39.1% (red) and associativity solves 16.9% (blue). Model performance on hold-out sequences is shown in black. **(c)** The model achieves high accuracy on almost all algorithmic distributions (97-100%), except for associative composition (60%).

element (i.e., $dc=c$) and applying the identity rule.

$$\text{``}kb = i, dc = c, cl = p, jp = l, \mathbf{dp = p}, en = e, bb = n, pj = l, dp = \text{''} \tag{2}$$

$$\text{``}kb = i, \mathbf{dc = c}, cl = p, jp = l, dp = p, en = e, bb = n, pj = l, dp = \text{''} \tag{3}$$

To disambiguate between potential mechanisms, we design five targeted data distributions to test specific algorithms that can solve algebra sequences when a corresponding set of facts is present in the context. We describe each data distribution and the hypothesized algorithm it tests below:

1. **Verbatim Copying ($\mathcal{D}_{\text{copy}}$).** This data tests whether the model copies exact facts from the context. We construct sequences $s \in \mathcal{D}_{\text{copy}}$ to contain at least one duplicate of the final fact.

2. **Commutative Copying ($\mathcal{D}_{\text{commute}}$).** In many groups, knowing that $ab=c$ often implies that $ba=c$. This data tests whether the model copies these commutative facts from the context. We construct $s \in \mathcal{D}_{\text{commute}}$ to contain at least one instance of the commutative fact (e.g., $yx=z$ for final fact $xy=$), and no duplicate facts.

3. **Identity Element Recognition ($\mathcal{D}_{\text{identity}}$).** This data tests whether the model can recognize and apply the identity rule. We construct $s \in \mathcal{D}_{\text{identity}}$ such that the final fact contains an identity element (e.g., $xy=x$) and that at least one prior fact in the context reveals the identity element (e.g., $zy=z$). We remove any duplicate or commutative facts.

4. **Associative Composition ($\mathcal{D}_{\text{associate}}$).** This data tests whether the model can chain fact results together to answer a new fact via associativity. Given a final fact $xy = z$, we construct $s \in \mathcal{D}_{\text{associate}}$ so that it contains a minimum set of facts that would enable a solution via association. For example, the three facts $xg=f$, $gd=y$, $fd=z$ can be composed (i.e., $(xg)d=fd \Rightarrow x(gd)=z \Rightarrow xy=z$) to compute $xy=z$. We make sure to only use triples without duplicate or commutative facts.

5. **Closure-Based Cancellation ($\mathcal{D}_{\text{cancel}}$).** This data tests whether the model can track group membership and appropriately apply the cancellation law to eliminate invalid answers (e.g., $xb=g$ eliminates $g$ as an answer to "$xy=$"). Given a final fact $xy=z$, we construct $s \in \mathcal{D}_{\text{cancel}}$ by including all the facts that share $x$ in the left-slot (e.g., $xb=g$) or $y$ in the right-slot (e.g., $cy=e$), and removing duplicate and commutative facts.

### 4.1 Measuring Coverage and Performance on Targeted Distributions

We seek to answer two questions: (1) What fraction of in-context algebra sequences can theoretically be solved by these hypothesized algorithms? and (2) Does the model successfully solve sequences that algorithmic strategies can solve when presented with the appropriate facts in-context?

**Algorithmic Coverage.** To understand the breadth of data that our hypothesized mechanisms might explain, we implement Python equivalents of all five algorithms (Appendix G) and measure their *coverage*, i.e., the percentage of sequences they can theoretically solve. We apply the algorithms

sequentially in the following order: verbatim copying, commutative copying, identity recognition, closure-based cancellation, and associative composition, where each algorithm is only applied to sequences unsolved by previous mechanisms. We compute algorithmic coverage over both random training sequences (Figure 3a) and random hold-out sequences where neither form of copying is possible (Figure 3b), using 2000 sequences for each evaluation.

We find that verbatim copying can solve a large percentage of the training data, with its area under the curve (AUC) being $67.9\%$ (Figure 3a, green). Commutative copying accounts for an additional $12.1\%$ of cases (purple), with the identity solving $4.2\%$ (yellow), closure-based cancellation solving $2.7\%$ (red), and associativity solving $3.6\%$ (blue) for total coverage AUC of $90.4\%$. In contrast, the model accuracy (black) achieves an AUC of $92.4\%$. While the hypothesized algorithms can explain most of the model's empirical training performance, they do not explain everything the model has learned ($\sim 2.0\%$ AUC, gray); there may be other interesting mechanisms this analysis misses.

When a sequence cannot be solved via copying or commutative copying, we see a very different trend (Figure 3b). In this more challenging setting, the model achieves a slightly lower AUC of $87.3\%$ (black). Identity recognition is able to solve $28.7\%$ of hold-out cases (yellow), closure-based cancellation can solve another $39.1\%$ (red), and associativity solves $16.9\%$ (blue) bringing the total hold-out coverage AUC to $84.7\%$. Here, the AUC gap between the model's empirical performance and our algorithmic coverage is $2.6\%$ (gray), and is primarily for algebra sequences with fewer facts.

**Model Performance on Subdistributions.** We evaluate the model on 400 sequences sampled from each distribution $\mathcal{D}_i$, and report results at $k = 50$ and $k = 100$ facts (Figure 3c), with more results in Appendix C. We find the model gets near perfect performance on four of the five data distributions that we test: verbatim copying ($100.0\%$), commutative copying ($99.0\%$), identity element recognition ($100.0\%$), and closure-based cancellation ($97\%$). However, model performance on sequences that test associative composition is worse ($60.2\%$), suggesting only partial learning of this property.

## 5  CAUSAL VERIFICATION OF LEARNED MECHANISMS

Based on the results in Section 4.1, we perform causal interventions to understand how the model mechanistically implements the algorithms with stronger empirical evidence: (1) verbatim copying, (2) commutative copying, (3) identity element recognition, and (4) closure-based cancellation.

### 5.1  CAUSAL INTERVENTIONS

In order to understand the internal computations underlying the model's capabilities, we use causal interventions (Vig et al., 2020; Meng et al., 2022; Geiger et al., 2025) to verify how the model implements the targeted behavior. This is typically done by implicating model components such as attention heads or directions in a model's activation space (Wang et al., 2023; Geiger et al., 2024; Mueller et al., 2025). Similar to prior work, we quantify the importance of a component via its indirect effect (IE; Pearl, 2001). We compute IE as the change in probability of the target variable token $v_{\text{target}}$ under some intervention across a pair of algebra sequences that differ in a meaningful way ($s_{\text{clean}}, s_{\text{corrupt}}$). Equation 4 shows an example of computing IE for an attention head $a^{(l,h)}$ at layer $l$, head $h$, by patching its activations from $s_{\text{clean}}$ into $s_{\text{corrupt}}$:

$$\text{IE}(l, h) = P(v_{\text{target}} | a^{(l,h)}_{s_{\text{clean}}} \to s_{\text{corrupt}}) - P(v_{\text{target}} | s_{\text{corrupt}}) \tag{4}$$

where $a^{(l,h)}_{s_{\text{clean}}} \to s_{\text{corrupt}}$ indicates activations $a^{(l,h)}$ are being patched (or replaced) from $s_{\text{clean}}$ into the same location in $s_{\text{corrupt}}$. The average indirect effect (AIE) can be computed over a dataset $\mathcal{D}$ as:

$$\text{AIE}(\mathcal{D}, l, h) = \frac{1}{|\mathcal{D}|} \sum_{\mathcal{D}} (\text{IE}(l, h)) \tag{5}$$

### 5.2  COPYING AND COMMUTATIVE COPYING

In this subsection, we investigate how the model implements verbatim and commutative copying (see also Appendix D). As shown in Section 4.1, a large percentage of our training data ($\sim 80\%$) can either be solved by verbatim copying or commutative copying (Figure 3a), and the model achieves high performance ($97$-$100\%$) when either form of copying is possible (see Figure 3c).

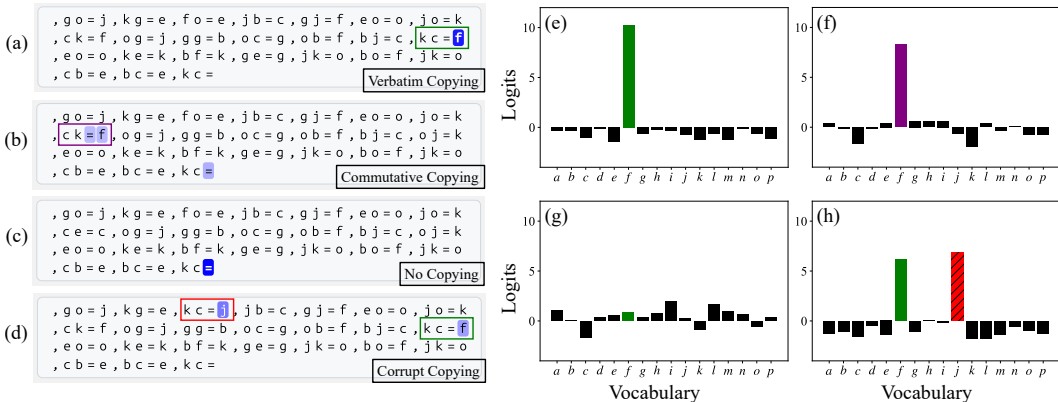

Figure 4: An analysis of copying (§ 5.2). Attention patterns (a-d) and direct logit contributions (e-h) of the copying head (layer 3, head 6) across variations of the same algebra sequence. (a) When verbatim copying is possible, the head attends to the answer-slot of the previous fact "$kc = f$" and (e) directly promotes that token's logit (green). (b) When the exact fact is absent, the head's attention shifts to the answer-slot and predictive token of the commutative fact "$ck = f$" and (f) promotes that token (purple). Note this fact was also in (a) but not attended to, indicating exact facts take precedence over commutative ones. (c) When both exact and commutative facts are absent, the head often self-attends and (g) no longer strongly promotes one token. (d) When injecting a matching "corrupted" fact with an incorrect answer ("$kc = j$", red), the head attends to each answer-slot and (h) promotes both variables (green, red).

**Verbatim Copying.** We search for attention heads responsible for copying the correct answer from the context by computing the average indirect effect (AIE) of each attention head $a^{(l,h)}$ (layer $l$, head $h$). We patch the activations of each head $a^{(l,h)}$ from the final predictive token in $s_{\text{clean}}$, taken from $\mathcal{D}_{\text{copy}}$, into the same token position in $s_{\text{corrupt}}$, a randomly sampled sequence where copying is not possible, and measure its IE. We compute $\text{AIE}(\mathcal{D}_{\text{copy}}, l, h)$ over 200 samples from $\mathcal{D}_{\text{copy}}$.

We find a single attention head (layer 3, head 6) with high AIE (0.91) that is primarily responsible for copying, with no other head having an AIE higher than 0.08 (Figure 13a). We visualize the attention patterns of this copying head in Figure 4a, and find that it attends to the answer-slot of duplicated facts (shown in green), much like the n-gram heads observed in Akyürek et al. (2024). On $\mathcal{D}_{\text{copy}}$, head 3.6 strongly promotes the logit of the attended-to token (Figure 4e) which can be seen by applying the model's unembedding matrix to the attention head output $U(a^{(l,h)})$. This allows us to understand its output contribution in terms of vocabulary tokens (Nostalgebraist, 2020; Elhage et al., 2021; Dar et al., 2023). The top logit consistently matches the attended-to token (Figure 14).

**Commutative Copying.** In many groups, knowing that $ab = c$ will also imply that $ba = c$. To investigate how the model implements such commutative copying, we compute the indirect effects of patching attention head activations from sequences in $s_{\text{clean}} \in \mathcal{D}_{\text{commute}}$, to non-copying sequences $s_{\text{corrupt}}$, where neither verbatim nor commutative copying are possible. We find the same pattern: head 3.6 is again the only head with strong AIE (0.48) for commutative copying (Figure 13b). In the absence of duplicate facts, head 3.6 attends to the predictive token and answer-slot of the *commutative* fact (Figure 4b) and similarly promotes the attended-to token (Figure 4f).

**Non-Copying Sequences.** When neither copy-inducing fact is present in the context, head 3.6 often self-attends (Figures 4c), not strongly promoting any token (Figure 4g). However, when the query contains an identity fact, we find head 3.6 has an interesting identity demotion behavior (§ 5.3).

**Corrupt Copying.** While copying the answer-slot of a commutative fact can solve facts for abelian groups, this doesn't work for non-commutative facts. When analyzing the copying behavior of head 3.6 on cyclic and dihedral groups separately, we find that more than 97% of the time it promotes the token it attends to, even if that token is the *wrong answer* (see Figure 14b). We illustrate this behavior in Figure 4d, where we inject a duplicate fact with an incorrect answer and show that head 3.6 attends to both duplicates (red, green) and promotes both of their logits (Figure 4h).

Figure 5: Identity Recognition. (a) PCA decomposition of fact hidden states at the final attention layer reveals a clear separation of identity facts (blue) and non-identity facts (red). (b) Head 3.1 promotes the logits of both variables in the query ($a$ and $e$), while head 3.6 demotes the logit of the identity variable, $e$. (c) PCA steering on its own can induce identity behavior, but it promotes both variables in the query to have near-equal logits. Inserting a false identity fact for either query variable triggers identity demotion, which, along with PCA steering, achieves cleaner identity control.

## 5.3 IDENTITY RECOGNITION

Our coverage analysis has revealed that when verbatim and commutative copying are no longer allowed, the identity algorithm can solve close to $30\%$ of all hold-out problems. In this section, we use data from $\mathcal{D}_{\text{identity}}$ to study how the model solves sequences where the query is an identity fact. Recall that an identity element $e \in G$ satisfies $e \cdot x = x \cdot e = x$ for all elements $x \in G$, so that if one variable in the question is known to be the identity, the answer is equal to the other variable.

Our experiments suggest that identity recognition emerges from the interaction of two complementary mechanisms: query promotion, that elevates both variables in the question as potential answers, and identity demotion that suppresses the known identity element. When both mechanisms activate simultaneously, the non-identity token is correctly selected.

**Structure from PCA.** First, we note that our transformer's representations reveal a strong signal correlated with the presence of an identity element in the question. To analyze this, we use PCA to plot final-layer attention outputs at the predictive token position (the "=" symbol) just before the model predicts an answer. There is a clear separation between facts containing identity elements (blue) and non-identity facts (red) along the first PCA dimension (Figure 5a). This separation is invariant to the specific variables in the fact or the underlying group. This suggests the model has learned to recognize and solve identity facts differently from those without an identity element.

**Query Promotion and Identity Demotion.** To analyze the role of the final layer attention at predicting identity facts in Figure 5b we use the logit lens (Nostalgebraist, 2020) and find two heads whose logits correlate strongly with identity variables. Head 3.1 promotes both variables in a given fact, serving as a "query promotion" mechanism. This strategy of predicting that the answer is equal to the question is appropriate for problems in which one of the factors is the identity, although on its own it would have the undesirable effect of promoting the identity element itself as the answer.

On identity fact sequences, head 3.6 acts as an "identity demotion" mechanism, attending to previous identity facts in the context and suppressing the identity token's logit (Figure 5b, pink). Combined with the previous strategy, this serves to leave only the non-identity factor as the promoted answer.

**Causal Verification.** Our experiments suggest that the dominant PCA direction in representation space controls the query promotion submechanism. To understand the causal effects, we perform representation steering experiments along this learned direction (see Appendix E for details). When we intervene on the final layer attention output of a non-identity fact and steer it toward the identity cluster, the model begins producing equal logits for both query tokens (Figure 5c: i vs. ii).

In addition to query promotion, we can also manipulate the model's identity recognition by introducing false identity facts to influence the identity demotion signal. When we inject a fact incorrectly suggesting one of the query tokens is an identity element, the identity demotion head (3.6) responds by suppressing that token and causes a cleaner identity prediction (Figure 5c, iii, iv).

On the other hand, when the model is presented with a false identity fact in-context while the query is a non-identity fact, it typically confuses the prediction. However, if we steer in the negative PCA

direction (away from the identity cluster), the model recovers and correctly predicts the non-identity answer. These steering experiments demonstrate that the learned PCA direction has causal influence over the model's identity reasoning, enabling us to both induce and suppress identity predictions.

## 5.4 CLOSURE-BASED CANCELLATION

The closure-based cancellation algorithm is a combination of two key submechanisms: (i) tracking which variables belong to the same group (i.e., the *closure*), and (ii) systematically eliminating invalid answers using the *cancellation law*, which implies that for elements $x, y, z \in G$, if $y \neq z$ then $xy \neq xz$ and $yx \neq zx$ (see Appendix A).

We hypothesize the algorithm can be understood at a high level as computing the difference of two sets: $S_{\text{closure}}$ - $S_{\text{cancel}}$. Consider the sequence sampled from $\mathcal{D}_{\text{cancel}}$ shown in Equation 6. For the final query $pe =$, the closure contains all elements that have previously appeared in facts involving $p$ or $e$, i.e., $S_{\text{closure}} = \{p, e, f, a, n\}$. The cancellation law then eliminates candidates from facts that share a variable in the left- or right-slot: $p$ (from $pf = p$), $n$ (from $ee = n$), $f$ (from $ae = f$), and $e$ (from $pp = e$), leaving $a$ as the only valid answer (i.e., $S_{\text{cancel}} = \{p, n, f, e\}$).

$$\text{``}pf = p, ee = n, pf = p, pf = p, ae = f, pp = e, pf = p, pn = f, pp = e, pe = \text{''} \tag{6}$$

We use causal interventions to determine how the model implements these two submechanisms. Our analysis reveals evidence of both a closure subspace, that promotes the logits of variables in the same group, and an elimination subspace that demotes answers based on facts present in the context.

**Closure Submechanism.** The closure submechanism emerges naturally from autoregressive training: when predicting the right-slot of a fact like $xy =$, the model must identify which variables could plausibly follow $x$. These are precisely the elements that belong to the same group (i.e., the closure). In fact, when we analyze the model's predictions at left-slot positions, we find nearly uniform logits across all elements previously associated with that variable, confirming the model has learned how to compute group closure (see Figure 15, Appendix F).

Inspired by previous work showing subspaces can encode high-level causal variables (Geiger et al., 2024; Prakash et al., 2025), we aim to identify a subspace $W$ that captures the model's representation of the closure set, $S_{\text{closure}}$. We construct counterfactual pairs $(s, s')$ from $\mathcal{D}_{\text{cancel}}$ that have different closure and elimination sets $(S_i, S'_i)$ such that under intervention, the expected counterfactual answer corresponds to a modified set difference: $v_{\text{CF}} = S_{\text{closure}} - S'_{\text{cancel}}$, where the closure set comes from $s$ and the elimination set comes from $s'$. We perform subspace-level patching from $s$ into $s'$ and train $W$ to maximize the likelihood of producing the expected counterfactual output $v_{\text{CF}}$ (Equation 7). If the intervention causes the model to predict $v_{\text{CF}}$, we take this as evidence that the subspace $W$ correctly represents the hypothesized closure set.

$$P(v_{\text{CF}}|(Wa_s^l + (I - W)a_{s'}^l) \to a_{s'}^l) \tag{7}$$

We can measure its accuracy as how often the model's predicted answer under intervention matches the expected counterfactual target (Figure 17). We train a 16-dimensional $W$ on the model's final layer attention output $a^l$, and find that it can achieve good intervention accuracy (99.8%) after a few epochs of training on 2000 data pairs (Figure 18). More details are in Appendix F.2.

For the closure subspace, we train probes (Alain & Bengio, 2017; Belinkov, 2022) to understand what the subspace has learned, and how it represents variables. We train each probe on the subspace to detect whether a variable is in the group closure or not. We find probes are able to identify when a variable is in the closure subspace with high accuracy (Figure 20), and that these variable-level probes partially align with the model's unembedding matrix (Figure 21), furthering evidence that the closure subspace promotes variables it has seen before in the context.

**Cancellation Submechanism.** To understand the cancellation submechanism, we train a subspace using a similar construction to the above, but vary the patching setup. If this new subspace $W'$ captures the elimination set, then it should generate the counterfactual answer arising from the opposite set difference, where the closure comes from the corrupt sequence $s'$, and the elimination set comes from the clean sequence $s$, giving: $(S'_{\text{closure}} - S_{\text{cancel}}) = v'_{\text{CF}}$.

We similarly train this subspace and find it also achieves high intervention accuracy, indicating it successfully represents the elimination set (Figure 19). Intervening in this subspace causes variables to be less likely to be predicted by the model. Both subspaces are trained on a single attention layer, and we find they capture partial contributions from several attention heads (Appendix F.1).

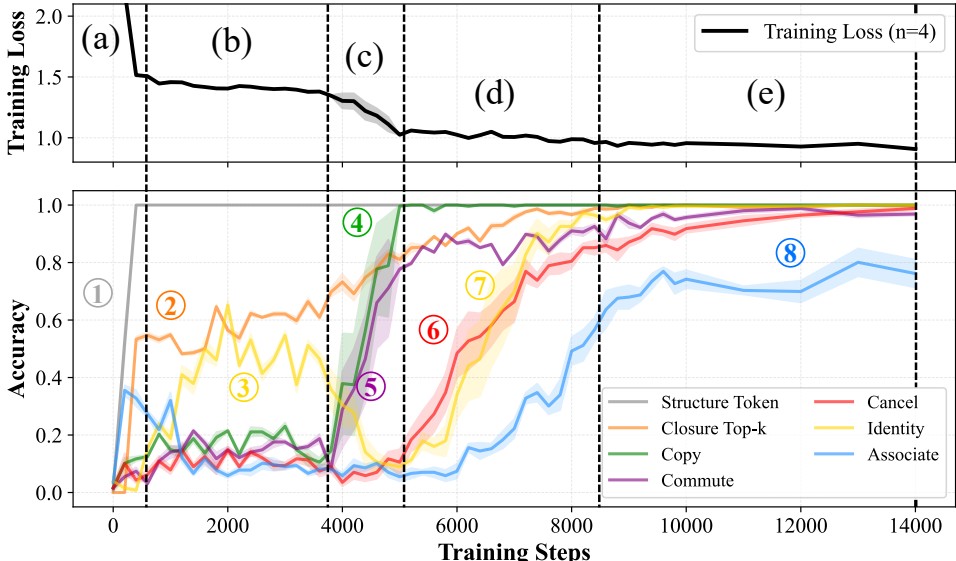

Figure 6: Dissecting Phase Transitions. (top) Average training loss of transformer models broken into 5 stages of learning. (bottom) We track 7 metrics corresponding to different skills the model acquires throughout training. (a) The first sharp drop in loss corresponds to learning to predict structural tokens: '=' and ',' (①, gray). (b) Next, the model begins to learn how to predict group closure (②, orange), and also learns the identity's query promotion submechanism (③, yellow, § 5.3). (c) This sharp drop in loss corresponds to the model learning how to copy answers verbatim from the context (④, green), along with commutative copying (⑤, purple). (d) After learning to copy, the model quickly improves on cancellation (⑥, red) and identity sequences (⑦, yellow) in parallel. We hypothesize this joint improvement corresponds to the fact that both tasks share a similar "demotion" submechanism – identity demotion and cancellation. These complement the closure and query promotion submechanisms learned previously. (e) Accuracy on associative sequences increases last (⑧, blue), after all other mechanisms have been learned.

## 6    PHASE TRANSITIONS CORRESPOND TO LEARNING OF DISCRETE SKILLS

We find that models undergo distinct phase transitions during training (Figure 6; see also Appendix C). Across seeds and configurations, the same sequence of stages marked by drops or plateaus in loss recurs (Figure 7). We study the training loss by computing several metrics at each model checkpoint. For each hypothesized mechanism (§4), we evaluate the model using 128 randomly sampled data points from the corresponding datasets described in Section 4. For structural tokens, we compute the model's accuracy of predicting '=' and ',' tokens across a batch of 128 prompts. For computing group closure, we measure the top-K matching accuracy at the left-slot position (more details are provided in Appendix F).

The earliest ability to emerge is **prediction of structural tokens**: '=' and ',' (Figure 6a, gray). This is followed closely by **group closure** (§ 5.4): the model learns that combining two elements always yields another valid group element (Figure 6b, orange). This ability appears in left-slot predictions of facts, where the model distributes probability nearly uniformly across all valid candidates (Appendix F). At the same time, the model learns the query promotion submechanism (Figure 6b, yellow), achieving around 50% on identity sequences (§5.3). The next sharp drop in loss corresponds to the model learning **contextual copying** (§ 5.2), first reproducing facts verbatim (Figure 6c, green) and then extending to commutative copying (Figure 6c, purple).

Later mechanisms emerge more gradually. The model develops **identity recognition**, steadily improving on identity-related facts and acquires **elimination reasoning** in parallel, applying cancellation laws and closure constraints to rule out inconsistent candidates. Unlike closure and copying, these abilities do not show sharp transitions but appear jointly, suggesting they build on top of copying: once the model can retrieve and recombine facts, it can also infer identities and apply elimination strategies (Appendix B). We hypothesize these are learned jointly because the identity demotion

mechanism (§5.3) and the elimination subspace (§5.4) perform similar functions, and their "promotion" submechanism counterparts are learned at similar times earlier in training. Finally, models begin to solve some associative sequences, after all other mechanisms have been learned.

# 7 RELATED WORK

**Arithmetic as a testbed for interpretability.** Arithmetic tasks have long served as controlled settings for studying and interpreting transformers (Liu et al., 2023). Small transformers trained on modular arithmetic exhibit "grokking" where they first memorize training data before converging to interpretable, generalizing solutions with periodic embeddings (Power et al., 2022; Liu et al., 2022; Nanda et al., 2023; Zhong et al., 2023; Stander et al., 2024; Morwani et al., 2024). Pretrained LLMs exhibit similar periodic structure in their number embeddings (Zhou et al., 2024; Hu et al., 2025; Kantamneni & Tegmark, 2025; Nikankin et al., 2025), enabling modular arithmetic without explicit training. Deng et al. (2026) find that arithmetic-fine-tuned LMs rely on symbolic subgroup patterns, instead of using partial products, but Bai et al. (2025) show that implicit chain-of-thought training *does* induce partial products and Fourier number representations. More closely related to our setting, He et al. (2024) show that transformers trained on permutations of one group develop hierarchical "circle-of-circles" representations, and Zhong & Andreas (2024) demonstrate that models with trained embeddings, but otherwise frozen random weights can still implement familiar geometric solutions. While these works study arithmetic settings where tokens have some fixed structure, our work examines a complementary setting where we remove fixed meanings of tokens altogether, requiring models to solve problems where token referents vary arbitrarily between sequences.

**Mechanisms of in-context learning.** The ability of transformers to learn from demonstrations has been attributed to several mechanisms. Early work identifies *induction heads* that underlie copying (Elhage et al., 2021; Olsson et al., 2022; Feucht et al., 2025), while theory frames ICL as Bayesian inference (Xie et al., 2022; Akyürek et al., 2023; Wurgaft et al., 2025) or gradient-descent-like adaptation (Von Oswald et al., 2023). More recent studies show LM representations capture task-level structure (Todd et al., 2024; Hendel et al., 2023; Yin & Steinhardt, 2025; Minegishi et al., 2025), and token representations flexibly adapt to context (Park et al., 2025a; Marjieh et al., 2025).

**Symbolic reasoning and causal interpretability.** Neural systems have long been studied as potential mechanisms for symbol manipulation, from tensor product (Smolensky, 1990) and holographic reduced representations (Plate, 1995) to recent cognitive-science studies of emergent symbolic reasoning in modern networks (Swaminathan et al., 2023; Yang et al., 2025). More recently, mechanistic interpretability has started mapping internal symbolic reasoning circuits in transformers (Li et al., 2023; Brinkmann et al., 2024; Prakash et al., 2024; Saparov et al., 2025; Wu et al., 2025; Li et al., 2025), using causal intervention techniques (Mueller et al., 2025; Geiger et al., 2024; 2025).

**Variables versus value processing in LMs.** A few works have tried to disentangle the ability of LMs to solve math abstractly from their ability to perform arithmetic computation. Cheng et al. (2025) find that LMs are better at abstract variable-based formulation of solutions compared to numeric computation of the same word problems, while Calais et al. (2025) find that in other problem settings LMs struggle with textual comprehension more than equation solving. Mirzadeh et al. (2025) similarly find that LMs lack robustness to changes in numeric values of math problems.

# 8 CONCLUSION

We have studied LMs trained on a focused algebra task designed to isolate abstract in-context reasoning behavior in the absence of fixed-meaning symbols. Our findings suggest that the kinds of reasoning strategies learned by transformers are dependent on the task structure. In our in-context algebra setting, where tokens carry no fixed meaning, we have analyzed the mechanisms learned by transformer LMs in detail and found that models develop symbolic mechanisms instead of the familiar parametric or geometric strategies found in settings where tokens *do* have fixed meanings. We have seen that transformers can learn to manipulate symbols in-context *without* needing to refer to their underlying meaning, similar to the way that high-school algebra students learn to solve math problems by manipulating letter variables without constantly thinking about the values they might contain (Usiskin, 1988). Understanding when and why models choose different computational strategies remains an important open question for future interpretability work.

## ETHICS STATEMENT

This paper aims to advance the foundational understanding of in-context learning and transformers. While such research may influence future model development and deployment, we cannot meaningfully anticipate these downstream impacts within the scope of this work.

## ACKNOWLEDGMENTS

The authors would like to thank Chris Wendler, Natalie Shapira, Nikhil Prakash, Arnab Sen Sharma, Freya Behrens, and Dana Arad for advice on various stages of the project and also Grace Proebsting and Michael Ripa for feedback on the paper. We are grateful for the generous support of Coefficient Giving (ET, RG, DB) and the National Science Foundation (Grant No. 2403303; RG, DB).

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

## A    GROUP THEORY

In this section, we review relevant terms from group theory that are used in our analysis.

A **group** $(G, \cdot)$ is a non-empty set $G$ equipped with a binary operation $\cdot : G \times G \to G$ that satisfies the following properties:

- **Associativity.** For all elements $x, y, z \in G$: $(x \cdot y) \cdot z = x \cdot (y \cdot z)$
- **Identity.** There exists an identity element $e \in G$ such that $e \cdot g = g \cdot e = g$ for all $g \in G$
- **Invertibility.** For each element $g \in G$, there exists an inverse element $g^{-1} \in G$ such that $g \cdot g^{-1} = g^{-1} \cdot g = e$

The **order** of a group is the number of elements contained in the set $G$, and we denote it as $|G|$. For notational convenience, we refer to a group $(G, \cdot)$ simply as $G$.

A group $G$ is called **abelian** (or **commutative**) if for all $x, y \in G$, the following holds: $x \cdot y = y \cdot x$.

Our training data consists of two main families of groups: cyclic groups and dihedral groups (§ 2).

A **cyclic group** of order $n$, denoted $C_n$, consists of all powers of a single generator element: $C_n = \{e, g, g^2, \ldots, g^{n-1}\}$ where $g^n = e$ is the identity. Every cyclic group $C_n$ has the same structure as (i.e., is isomorphic to) doing arithmetic modulo $n$. For example, in $C_5$, multiplying group elements (e.g., Equation 8) works exactly like adding numbers mod 5 (e.g., Equation 9):

$$g^3 \cdot g^4 = g^2 \tag{8}$$
$$\equiv 3 + 4 = 2 \pmod 5 \tag{9}$$

A **dihedral group** $D_n$ is the group of symmetries of a regular $n$-gon (square, hexagon, etc.), with order $|D_n| = 2n$. Its elements consist of $n$ rotations and $n$ reflections. We note that while all cyclic groups are abelian, dihedral groups are non-abelian for $n \geq 3$.

An important consequence of group structure is the **cancellation law**, which states that we can "cancel" common terms in equations. Specifically, for any group $G$ with elements $x, y, z$:

- *Left cancellation:* If $xy = xz$, then $y = z$
- *Right cancellation:* If $yx = zx$, then $y = z$

Equivalently (by contrapositive): if $y \neq z$, then $xy \neq xz$ and $yx \neq zx$. This rule guarantees that distinct group elements produce distinct products which helps ground our understanding of the closure-based cancellation mechanism described in Section 5.4.

For completeness, we briefly describe other algebraic structures we test on that lack some subset of group properties:

- A **semigroup** is a set with an *associative* binary operation, but does not require an identity element or inverses.
- A **quasigroup** is a set where equations $ax = b$ and $ya = b$ always have unique solutions for any $a, b$, but the operation need not be associative. Finite quasigroups are equivalent to Latin squares (Jacobson & Matthews, 1996).
- A **magma** is simply a set equipped with a binary operation, with no other required structural properties.

## B    DISCUSSION: CONTEXTUAL REASONING IS RICHER THAN JUST COPYING

In this section, we provide additional context about how our findings relate to prior work investigating the mechanisms underlying in-context learning.

Since the introduction of in-context learning (ICL; Brown et al., 2020), understanding the ICL capabilities of LLMs has become a major area of research (Dong et al., 2024; Lampinen et al., 2025). Seminal work by Elhage et al. (2021) and Olsson et al. (2022) studied ICL through the lens of

context-dependent loss reduction and identified *induction heads* that copy tokens from earlier in the context. Beyond literal copying, Olsson et al. (2022) hypothesized that much of in-context learning could be explained as "in-context nearest neighbors" or "analogical sequence copying" via induction heads that operate on higher-level abstractions beyond literal tokens.

Subsequent research has revealed increasingly sophisticated variations of induction-like behavior showing models transport abstract contextual information such as multi-token concepts (Feucht et al., 2025), few-shot task representations (i.e., function vectors) (Todd et al., 2024; Hendel et al., 2023; Yin & Steinhardt, 2025), syntactic and semantic relations (Ren et al., 2024), or abstract variables (Yang et al., 2025). At the implementation level, these mechanisms all involve attention heads moving information between token positions. However, each mechanism differs substantially in the kind of information they transport and how it is processed.

Thus, it is unclear from prior work whether all forms of contextual learning are best characterized as some form of "fuzzy copying" — where characterizing them by the level of abstraction at which they operate is enough — or if qualitatively different computational strategies emerge. For instance, when studying how LLMs perform tasks that require entity binding, several prior works have observed models transport token positional information pertaining to where tokens appear in a sequence (Feng & Steinhardt, 2024; Dai et al., 2024; Prakash et al., 2024; 2025; Wu et al., 2025). While it is still "copied" forward via attention, this form of contextual reasoning seems fairly distinct from traditional or "fuzzy" induction as it is related to a token's relative position instead of its meaning.

Studying this question is challenging in pretrained LLMs because token hidden states often reflect both immediate context and parametric knowledge encoded during pretraining. This entanglement makes it difficult to isolate pure contextual reasoning. To study this in a more controlled setting, we design a contextual reasoning setting that eliminates parametric knowledge by removing fixed token meanings entirely. Our in-context algebra task assigns tokens to algebraic elements randomly within each sequence, forcing models to infer all structure from observed relationships alone (§ 2).

We analyze the mechanisms that transformers consistently develop when trained on our in-context algebra setting and find that they still learn both induction-style copying and "fuzzy" commutative copying (§5.2). This is expected, as copying can solve a large portion of the training data on its own (§4.1, Figure 3a). However, we also identify two additional mechanisms that seem qualitatively different from the fuzzy copying strategies seen in prior work: identity element recognition, and closure-based cancellation. At the level of individual operations, these are not entirely unprecedented; inhibition signals or demotion of tokens through attention heads have been documented before (Wang et al., 2023; Merullo et al., 2024), and closure-based cancellation relies on a similar elimination submechanism.

What seems distinctive is not the operations themselves but what they operate over: rather than suppressing based on fixed syntactic or semantic token information, both mechanisms are driven by relational structure derived entirely in-context. Identity recognition works by observing which element leaves others unchanged across multiple facts, then applying both query promotion and a conditional suppression rule to select the non-identity variable (see §5.3 and Appendix E). Closure-based cancellation infers group membership to construct a set of valid candidates, then systematically eliminates those ruled out by the cancellation law (§5.4, and Appendix F).

These mechanisms are examples of how models can reason about tokens as variables by manipulating references in-context without knowing their underlying meaning. Taken together, our findings provide additional evidence that the contextual strategies that transformers can develop look increasingly different from mere copying. Understanding when and why models learn different contextual strategies remains an important open question for future work.

## C   MODEL ARCHITECTURE AND TRAINING

In this section, we provide more details about our training setup.

**Model Training Details.** We train autoregressive transformer models (Vaswani et al., 2017), on in-context algebra sequence data sampled as described in Section 2, with a batch size of 128, and sequences with 200 facts. Our vocabulary consists of 16 variable tokens, a predictive token, and a separator token ($N = 18$ in total), and each fact is made up of 5 tokens. As an additional hold-out

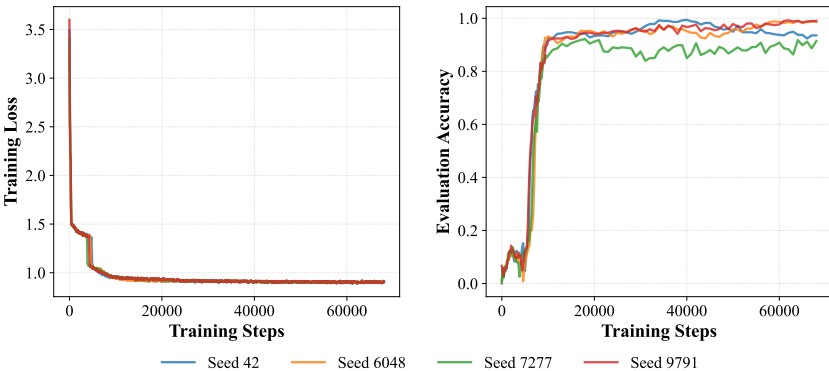

Figure 7: **Consistency across runs.** We observe qualitatively similar patterns across multiple training runs, both in terms of (a) phase transitions in the loss curves and (b) corresponding "grokking" increases in hold-out evaluation performance.

scheme, we select one variable token (e.g., $a$) that is never assigned as an identity element during training. This ensures that we can test both per-sequence hold-outs (where copying is not possible), and variable-assignment hold-outs where the exact assignment of elements to variables is unseen.

**Tooling and Compute.** Our transformer implementation is based on code adapted from nanoGPT (Karpathy, 2022); we use Rotary Positional Embeddings (RoPE) (Su et al., 2024) instead of learned positional embeddings. We use the AdamW optimizer (Loshchilov & Hutter, 2019) with a learning rate of $10^{-5}$, and 1000 warmup steps. The primary model we study in the main paper has 4 layers, with 8 attention heads per layer, and a hidden state dimension of 1024. As shown in Figure 7, we observe qualitatively similar patterns across different seeds, both in terms of phase transitions in the loss and hold-out evaluation accuracy. We usually see convergence (eval. accuracy $\geq 99\%$) between 30,000 and 75,000 steps and save the checkpoint with the best evaluation accuracy.

We train our models with either NVIDIA A100 80GB GPUs or NVIDIA A6000 48GB GPUs. Training statistics are logged using Weights and Biases (Biewald, 2020). Experiments are implemented using NNsight (Fiotto-Kaufman et al., 2025) and PyTorch (Paszke et al., 2019), and run on workstations with NVIDIA A6000 48GB GPUs. We use SymPy (Meurer et al., 2017) for simulating various group structures for our in-context algebra setting and use a custom implementation for magmas, semigroups, and quasigroups, with quasigroup generation based on Jacobson & Matthews (1996)'s method for Latin squares.

### C.1 MODEL ARCHITECTURE HYPERPARAMETERS

In this subsection, we study the effect of three hyperparameters that govern model capacity: (1) number of layers, (2) number of attention heads per layer, and (3) hidden state dimension. The training loss and evaluation accuracy of each hyperparameter sweep is shown in Figure 8, and the corresponding breakdown of model performance by metric and hyperparameter configuration is shown in Table 1, where results are reported using the checkpoint with the highest evaluation accuracy.

For each hyperparameter setting, there are noticeable drops in loss throughout training which correspond to the learning of discrete skills relevant for the task (§ 6), and is consistent with similar findings of phase transitions and skill-learning in prior work (Olsson et al., 2022; Nanda et al., 2023; Singh et al., 2024; Chen et al., 2024; Kangaslahti et al., 2025; Kim et al., 2025; Hoogland et al., 2025). In general, models with more capacity (i.e., more layers, larger hidden dimension, or more heads) learn the task more quickly and have shorter loss plateaus. Having fewer layers corresponds to longer loss plateaus, and delayed generalization (Figure 8, left). Models with smaller model dimensions ($\sim d \leq 256$) fail to generalize well when trained on the same amount of data (Figure 8, middle). Using hidden size $d = 128$ achieves accuracy near random guessing and $d = 256$ only achieves $\sim 60\%$ accuracy while the models with hidden size $d \geq 512$ achieve $95\%$ evaluation accuracy or above. Models trained with only two attention heads per layer exhibit delayed generalization compared to four or eight heads per layer (Figure 8, right).

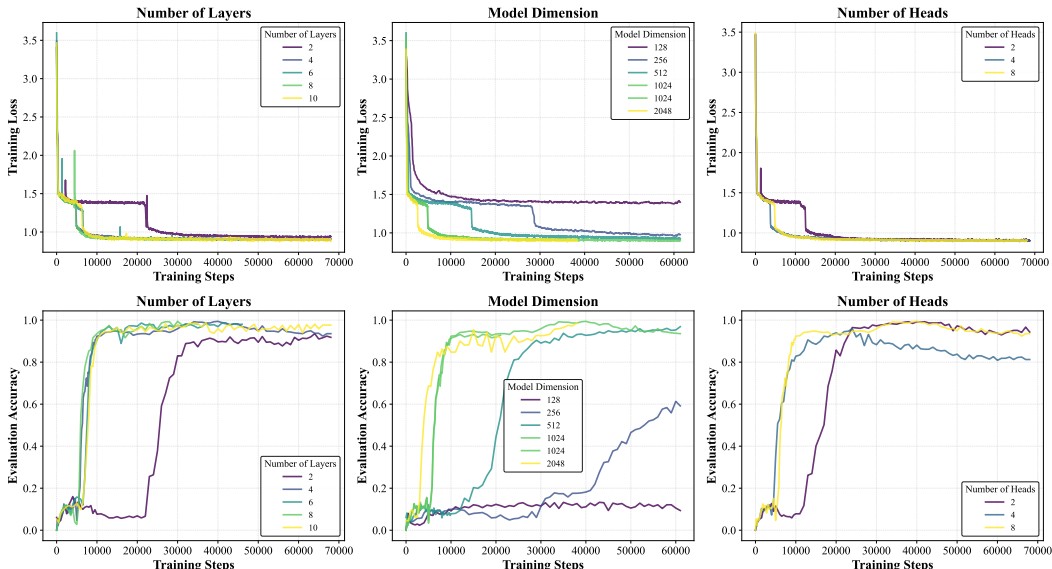

Figure 8: (top) Training loss over training steps for various architecture hyperparameters: number of layers, model hidden dimension, and number of attention heads per layer. (bottom) Evaluation accuracy for each hyperparameter sweep. With more model capacity (more layers, larger hidden size, or more heads), models achieve better performance and converge more quickly. There are some cases for model dimension where the model only partially converges (or does not converge at all), suggesting the model needs sufficient capacity to solve this task.

Table 1: Effect of model architecture hyperparameters: layers, heads, and hidden size. For each configuration, we report metrics at the training step with maximum evaluation accuracy (up to 60,000 steps). Scores under 90% are highlighted in red, indicating poor performance. Results show that models require sufficient hidden size (dimension $\geq$ 512) to learn the task effectively. In general, more capacity yields better evaluation performance. Associativity scores show high variance (and consistently lower scores) across all configurations despite consistent evaluation accuracy. Corresponding training curves are shown in Figure 8.

| Configuration | | | Evaluation Metrics | | | | | |
|---|---|---|---|---|---|---|---|---|
| # Layers | # Heads | Dim. | Eval. Acc. | Copy | Commute | Identity | Associativity | Closure |
| Sweep 1: Number of Layers | | | | | | | | |
| 2 | 8 | 1024 | 93.5% | 100.0% | 98.4% | 100.0% | 78.1% | 100.0% |
| 4 | 8 | 1024 | 99.4% | 100.0% | 100.0% | 100.0% | 85.9% | 100.0% |
| 6 | 8 | 1024 | 98.6% | 100.0% | 100.0% | 96.9% | 59.4% | 100.0% |
| 8 | 8 | 1024 | 99.4% | 100.0% | 100.0% | 100.0% | 51.6% | 100.0% |
| 10 | 8 | 1024 | 98.8% | 100.0% | 100.0% | 100.0% | 75.0% | 100.0% |
| Sweep 2: Number of Attention Heads | | | | | | | | |
| 4 | 2 | 1024 | 99.2% | 100.0% | 100.0% | 100.0% | 59.4% | 100.0% |
| 4 | 4 | 1024 | 95.9% | 100.0% | 100.0% | 98.4% | 64.1% | 100.0% |
| 4 | 8 | 1024 | 99.4% | 100.0% | 100.0% | 100.0% | 85.9% | 100.0% |
| Sweep 3: Hidden State Dimension | | | | | | | | |
| 4 | 8 | 128 | 13.3% | 9.4% | 17.2% | 62.5% | 17.2% | 51.6% |
| 4 | 8 | 256 | 61.3% | 100.0% | 87.5% | 85.9% | 56.2% | 97.7% |
| 4 | 8 | 512 | 96.9% | 100.0% | 95.3% | 100.0% | 82.8% | 100.0% |
| 4 | 8 | 1024 | 99.4% | 100.0% | 100.0% | 100.0% | 85.9% | 100.0% |
| 4 | 8 | 2048 | 98.6% | 100.0% | 100.0% | 100.0% | 87.5% | 100.0% |

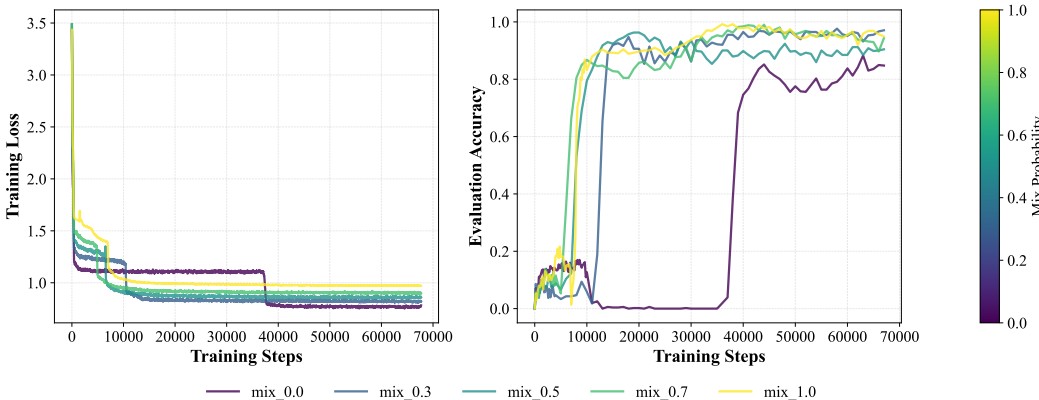

Figure 9: Effect of group sampling probability $p_{\text{mix}}$. We train five models with the same seed (42) and architecture (4 layers, 8 heads, 1024 hidden size), but vary the group sampling probability $p_{\text{mix}} \in \{0.0, 0.3, 0.5, 0.7, 1.0\}$. (a) The training loss curves for different values of $p_{\text{mix}}$ follow a consistent pattern: lower values of $p_{\text{mix}}$ have steeper early drops, but longer plateaus that follow. Higher values of $p_{\text{mix}}$ have shorter loss plateaus. (b) Evaluation accuracy for different values of $p_{\text{mix}}$. Training runs with higher values of $p_{\text{mix}}$ tend to achieve better held-out evaluation performance.

## C.2   GROUP SAMPLING PROBABILITY AND TASK DIVERSITY

To generate an in-context algebra sequence, we first sample a set of groups from the training distribution $\mathcal{G}$ (see Section 2). In this subsection, we provide more details about our sampling procedure, and investigate the effect of the group sampling probability $p_{\text{mix}}$ as a training hyperparameter.

When sampling a mixture of groups $\mathcal{G}_s$, for a sequence $s$, the first group is sampled uniformly from $\mathcal{G}$. After an initial group is chosen, additional groups are iteratively sampled with replacement with probability $p_{\text{mix}}$, continuing while the total order is less than or equal to the number of variables $|V| = N$ or a random draw from the interval $[0, 1]$ exceeds $p_{\text{mix}}$. A new group is added to $\mathcal{G}_s$ only if the total order of $\mathcal{G}_s$ would remain less than or equal to $N$. Thus, choosing $p_{\text{mix}} = 0$ results in sequences containing exactly one group, while $p_{\text{mix}} = 1$ maximizes the number of groups mixed within each sequence, up to total order $|V|$. Thus, $p_{\text{mix}}$ can be thought of as a measure of in-context task diversity. The algebra model we study in the main paper uses $p_{\text{mix}} = 0.7$.

In Figure 9a, we show loss curves and evaluation accuracy for transformer models trained with the same seed but different values of $p_{\text{mix}} \in \{0.0, 0.3, 0.5, 0.7, 1.0\}$. Training loss curves follow a consistent pattern: models trained with lower values of $p_{\text{mix}}$ have steeper early drops, but longer loss plateaus. Higher values of $p_{\text{mix}}$ correspond to shorter loss plateaus, but higher training loss. However, lower train loss does not necessarily correspond to higher evaluation accuracy (Figure 9b).

Recall that our evaluation data excludes copying and commutative copying sequences (Section 3). We find that models trained with higher values of $p_{\text{mix}}$ tend to achieve *better* held-out evaluation accuracy, even though they have higher training loss (Figure 9b). One reason for this might be that sequences generated using higher values of $p_{\text{mix}}$ have more groups per sequence, and thus more in-context task diversity. This aligns with findings from previous work showing that higher task diversity leads to more robust generalization (Raventos et al., 2023; Kirsch et al., 2024; Ye et al., 2024; Park et al., 2025b; Wurgaft et al., 2025). Similarly, since repetition is more likely to happen in sequences with fewer groups (i.e., lower values of $p_{\text{mix}}$), models trained with lower sampling probabilities have lower task diversity (e.g., copying is much more common as a possible solution).

An additional benefit of training with higher mixing probabilities (more groups per sequence) is that models tend to achieve high evaluation accuracy (generalize) *faster* than lower mixing probabilities. This was initially surprising, but is consistent with Kim et al. (2025) who show that increased task diversity actually shortens loss plateaus. While having more groups in a sequence is a more difficult problem, Figure 9b shows this configuration learns more quickly. Using a mixing probability of $p_{\text{mix}} = 0.0$, where only a single group is sampled per sequence, has the slowest time to held-out generalization, while higher values of $p_{\text{mix}}$ begin to generalize sooner.

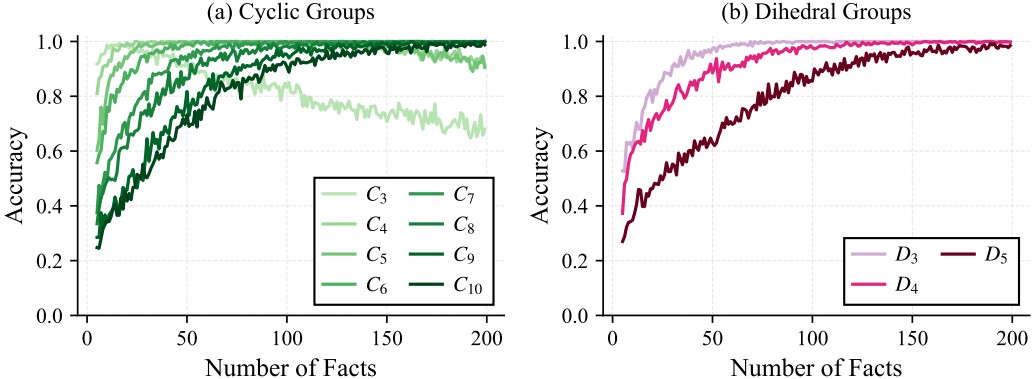

Figure 10: Heldout performance on in-distribution groups. (a) Model accuracy on cyclic groups generally increases with context length, except for very small groups which tend to degrade in performance with longer contexts. The model needs more facts to achieve the same performance with larger groups. (b) Dihedral groups follow a similar trend. Larger groups get better with more facts. $D_5$, which has 10 elements, reaches near-perfect accuracy around 200 facts, while smaller dihedral groups converge earlier (around 75-100 facts).

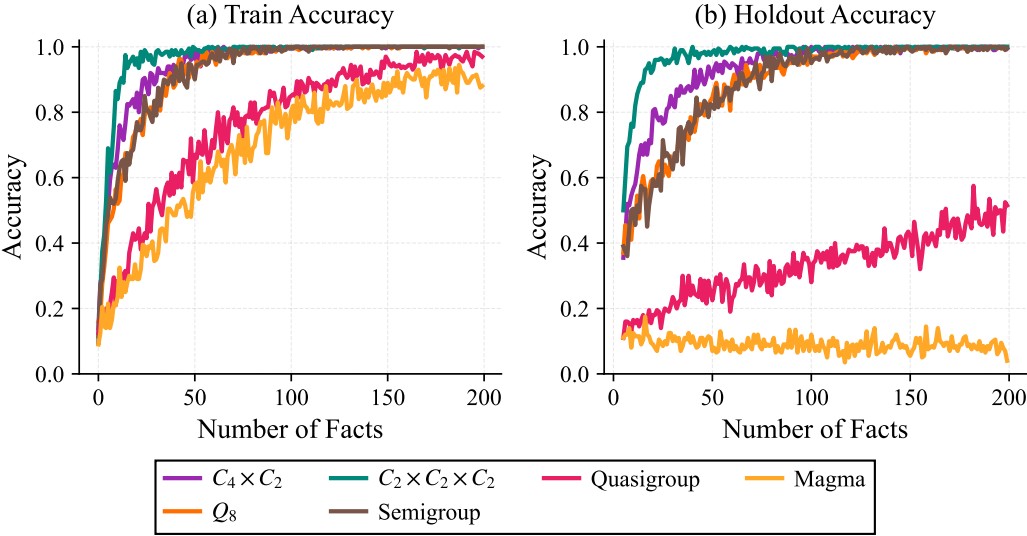

Figure 11: (a) Performance on algebraic structures unseen during training (where copying is possible). This includes the three unseen groups of order 8: $C_4 \times C_2$, $Q_8$, and $C_2 \times C_2 \times C_2$ (also called $\mathbb{Z}_2^3$), and 3 non-group structures: semigroup, quasigroup, and magma. The model achieves comparable performance on the unseen groups as it does to the in-distribution order 8 groups, while quasigroups and magmas have worse accuracy. (b) Model performance on held-out sequences for unseen algebraic structures. The hold-out performance of the model is surprisingly good for all groups, as well as the semigroup. However, holdout performance on the quasigroup is poor, only achieving a max of $50\%$ at 200 facts and the model performs even worse on the magma (near zero).

## C.3 PERFORMANCE ON GROUPS AND NON-GROUP ALGEBRAIC STRUCTURES

In this section, we compare the model's performance in-distribution to the model's performance on groups not seen during training, as well as non-group structures. Figure 10 shows the model's performance on in-distribution cyclic and dihedral groups. Performance typically increases with the number of facts in the sequence, and groups with more elements take longer to achieve perfect accuracy. For $C_3$, the performance actually begins to decreases after 25 facts.

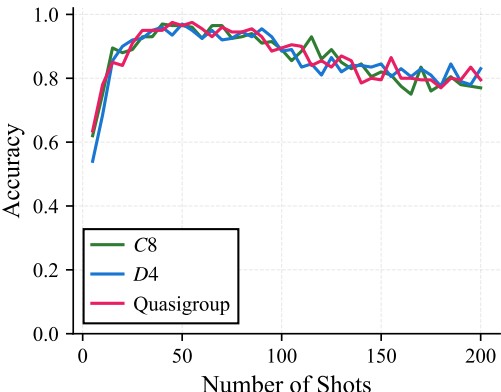

Figure 12: Model performance on a cancellation-based subset of quasigroup sequences. Although the overall hold-out performance of the model on quasigroups is poor ($\leq 50\%$) (see Fig. 11b), we find that performance is much better on quasigroup sequences where cancellation gives a unique answer, getting up to $100\%$ around 50 shots, and doing cancellation just as well as in-distribution groups. This suggests the closure-based cancellation mechanism learned by the model is a generalizable symbolic mechanism that does not depend on the specific algebraic structure of the data.

For unseen group structures of order 8, the model still performs very well (Figure 11). The holdout accuracy is similar to that of in-distribution groups. The model is also able to solve the semigroup with near-perfect accuracy with enough facts in the sequence, while hold-out performance (where copying is not possible) on quasigroups and magmas is significantly worse.

Finite quasigroups are naturally solvable via the cancellation law, thus we evaluate on a subset of quasigroup sequences where cancellation can solve the problem. We find that on this subset, the model does much better than the overall hold-out accuracy previously reported, providing evidence that some mechanisms (i.e., closure-based cancellation) learned by the model are generalizable symbolic mechanisms that do not depend on the specific algebraic structure the data is sampled from, as long as the data possesses that property.

## C.4 Additional Performance on Data Subsets

We extend Figure 3c to show performance on data subsets for varying number of facts (Table 2).

Table 2: Model performance on different data subsets from § 4. The model gets near-perfect accuracy ($97 - 100\%$) on almost all sequences, except for those solved via associativity, on which it maxes out at $65\%$ for 5-fact sequences.

| Key | Number of Facts | | | | | | | |
|---|---|---|---|---|---|---|---|---|
| | **5** | **10** | **25** | **50** | **75** | **100** | **150** | **200** |
| $\mathcal{D}_{\text{copy}}$ | 100.0% | 100.0% | 100.0% | 100.0% | 100.0% | 100.0% | 100.0% | 100.0% |
| $\mathcal{D}_{\text{commute}}$ | 92.0% | 89.0% | 95.0% | 98.0% | 98.0% | 99.0% | 100.0% | 99.0% |
| $\mathcal{D}_{\text{identity}}$ | 94.0% | 97.0% | 99.0% | 99.0% | 100.0% | 100.0% | 100.0% | 98.0% |
| $\mathcal{D}_{\text{associate}}$ | 65.0% | 62.0% | 66.0% | 60.2% | 62.0% | 56.5% | 50.0% | 40.0% |
| $\mathcal{D}_{\text{cancel}}$ | 57.0% | 75.0% | 94.0% | 97.0% | 94.0% | 92.0% | 81.0% | 76.0% |

## D Additional Results on Copying

In this section, we provide additional experimental details and results related to the copying and commutative copying mechanisms. Figure 13a shows a heatmap of the average causal effect of patching from verbatim copying sequences into no-copy sequences for each attention head in the

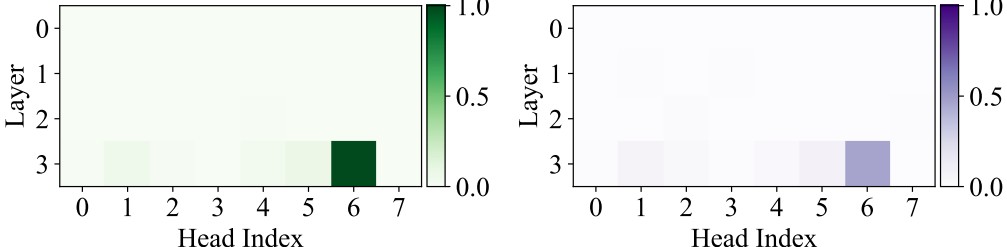

Figure 13: (a) Average causal indirect effect (Equation 5) for each attention head when patching from copying sequences into non-copying sequences, where darker green indicates a stronger change in probability. A single head (layer 3, head 6) is strongly implicated in verbatim copying behavior (AIE=0.91). (b) The same head is implicated when performing patching from commutative copying sequences into non-copying sequences, though the causal effect is slightly weaker (AIE=0.479).

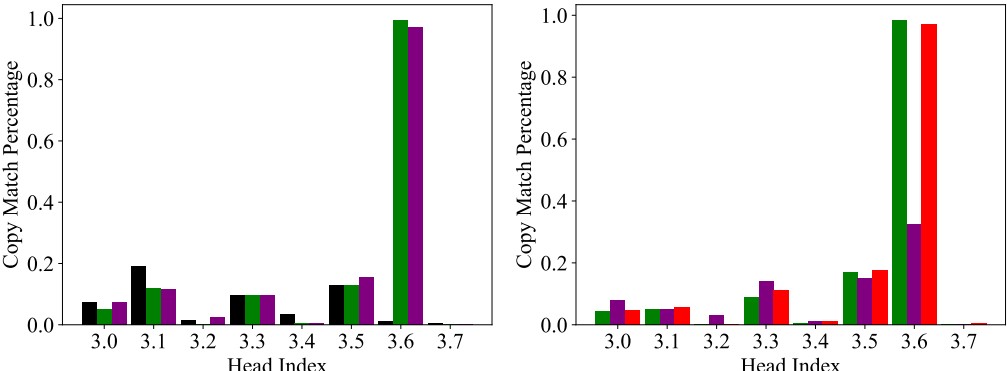

Figure 14: Decoding the output of each attention head at the final layer via the model's unembedding matrix reveals how often an attention head's highest logit matches the correct answer on copying sequences. (a) For cyclic groups, we see one head stand out: head 3.6's highest decoded logit matches the correct answer more than $99.5\%$ of the time on sequences where verbatim copying is possible (green), and $97\%$ of the time for commutative copying sequences (purple), while almost never promoting the correct answer on non-copying sequences (black). (b) For dihedral groups, where not all facts are commutative, we see a similar same trend for exact copying sequences (green, $\mathcal{D}_{\text{copy}}$), while for commutative copying head 3.6 only matches the correct answer $32.5\%$ of the time (purple, $\mathcal{D}_{\text{commute}}$). However, if we instead measure whether the highest decoded logit matches the *most attended-to token*, this happens $97\%$ of the time (shown in red).

model, computed as $\text{AIE}(\mathcal{D}_{\text{copy}}, l, h)$ for an attention head at layer $l$ and head index $h$. Figure 13b shows a similar heatmap of average causal effects for each attention head when patching from commutative copying sequences into no-copy sequences, similarly computed as $\text{AIE}(\mathcal{D}_{\text{commute}}, l, h)$ for each layer $l$ and head index $h$. In each case, we identify a single attention head (layer 3, head index 6), which has a much higher AIE than other heads and is primarily responsible for copying behavior.

To understand the behavior of these heads under various data settings, we characterize each head's output using direct logit attribution, or the "logit lens" (Nostalgebraist, 2020; Elhage et al., 2021). We apply the unembedding matrix to each attention head output (i.e., $U(a^{(l,h)})$ and compute how often the token with the highest decoded logit matches the target token.

Figure 14a shows how often each attention head promotes the correct answer token when using only cyclic groups to generate copying sequences. This is computed using 200 prompts for each of 3 prompt distributions: sequences where verbatim copying is possible ($s \in \mathcal{D}_{\text{copy}}$), sequences where commutative copying is possible ($s \in \mathcal{D}_{\text{commute}}$), and sequences where neither form of copying is possible. The highest logit promoted by the copying head (layer 3, head 6) almost always matches

the target answer for both verbatim (green) and commutative copying sequences (purple), but almost never on non-copying sequences (black).

However, performing this same analysis on sequences sampled using only dihedral groups yields a slightly different result (Figure 14b). When verbatim copying is possible, we still see head 3.6's top logit matches the correct answer token more than $99\%$ of the time (green), as expected. However, on sequences sampled from $\mathcal{D}_{\text{commute}}$, this value drops to $32.5\%$ (purple). If we instead measure whether the highest decoded logit matches the *most attended-to token*, this happens $97\%$ of the time for head 3.6 (shown in red). This is curious because for these sequences, head 3.6 seems to be "blindly" copying the symbol it attends to even though it is not the correct answer. While this strategy would solve any commutative pair of facts, it cannot solve non-commutative facts found in dihedral groups. In Figure 4d, we show a related behavior where head 3.6 will attend to and promote the answers of injected, incorrect facts in addition to correct ones.

## E   IDENTITY RECOGNITION DETAILS

In this section, we provide additional details about the steering experiments mentioned in Section 5.3. We use them study whether the separation of identity and non-identity facts seen via PCA in Figure 5a is causally relevant to downstream predictions.

To extract the PCA steering direction, we sample 1000 sequences, ensuring that each sequence has 200 facts, and gather final attention layer outputs at the predictive token position (the "=" symbol) for each sequence. We then perform PCA on the activations, and find that the variation along the first principle component separates identity facts from non-identity facts (Figure 5a). We denote this PCA direction as $\mathbf{v}_1$, and evaluate its role through several experiments. In each experiment we use a sample of 5000 non-identity sequences, each containing 200 facts.

**PCA Steering Controls Query Promotion.** We first test the effect of steering non-identity sequences towards the identity cluster along $\mathbf{v}_1$. Concretely, this is done for non-identity sequences by adding $5\mathbf{v}_1$ to the final attention layer output at the final prediction token. While the default behavior of the model is to predict an answer that is not present in the query fact, we find that steering towards the identity cluster using $\mathbf{v}_1$ causes the model to predict query variables $99.2\%$ of the time over the 5000 non-identity sequences we evaluate on. Upon examining the model's output logits, we find that both query variables are strongly promoted as likely answers (with near-equal logits). This suggests that $\mathbf{v}_1$ encodes a signal for "query promotion". Interestingly, query promotion by itself is a sufficient criteria to distinguish between identity and non-identity facts, since only identity facts will have one of the query symbols as an answer.

**Adding False Identity Facts Controls Identity Predictions.** While steering non-identity sequences with $\mathbf{v}_1$ can promote both query variables, it does not reliably promote one over the other. Recall that the other submechanism identified as important for solving identity facts was identity demotion. Next, we study whether we can cause the model to treat arbitrary non-identity facts as identity facts through a combination of PCA steering and injecting false identity facts into the context.

Concretely, given a non-identity sequence with a query $xy=$, we replace a random fact in the sequence with a false fact indicating that either the left-slot ($x$) or right-slot ($y$) should "act" as an identity element (e.g., $x?=?$, or $?y=?$, where '?' could be any variable). We then steer the model towards the identity cluster using $\mathbf{v}_1$ as before, and measure whether the model predicts the non-identity variable (e.g., $y$ or $x$) under this counterfactual setup.

We find that using PCA steering and injecting a false left-slot identity fact causes the model predict the right variable $96.1\%$ of the time. And similarly, PCA steering along with injecting a false right-slot identity fact causes the model to predict the left variable $79.6\%$ of the time. These results show that we can control whether a model treats a fact as an identity fact (i.e., uses its identity mechanisms) through a combination of two simple interventions: PCA steering along $\mathbf{v}_1$ to induce query promotion, and false identity fact injection to suppress a specific variable.

**Steering Away from the Identity Cluster.** Injecting a false identity fact into non-identity sequences confuses the model, causing it to wrongly predict one of the two query symbols. We find that steering away from the identity cluster (subtracting $\mathbf{v}_1$ instead of adding) can erase this confusion, causing the model to predict the correct non-identity answer $97.0\%$ of the time.

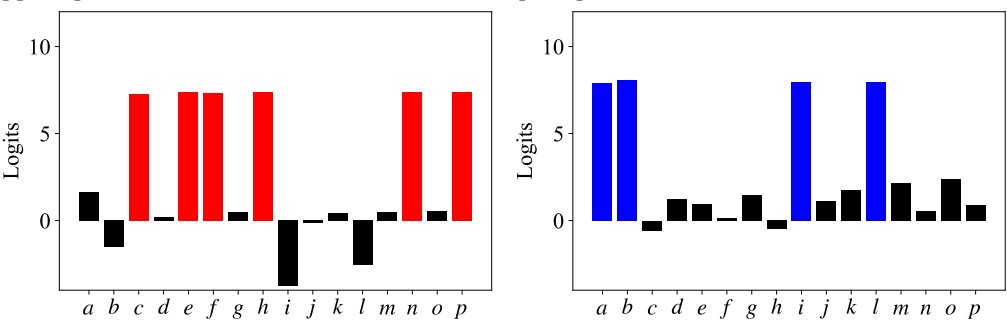

**Sequence:**`'hp=e,il=i,li=i,nc=c,bi=l,ne=e,fe=h,pp=f,ba=i,pc=h,eh=n,la=a, pp=f,pf=n,fe=h,fh=c,il=i,ef=h,hh=f,cp=h,pc=h,hh=f,cn=c,bi=l,`**(hc=/bi=)**`'`

Figure 15: Closure submechanism (§5.4). When predicting the right-slot of a fact, the model produces nearly uniform logits over all variables previously associated with the left-slot in the context. Here, we show the logits at the left-slot for the same sequence that differs only in the final query fact ($hc=$, vs. $bi=$). For different left-slot variables ($h$ vs. $b$), the model produces higher logits over either (a) the six elements connected to $h$: $\{c, e, f, h, n, p\}$ (shown in red), or (b) the four variables associated with $b$: $\{a, b, i, l\}$ (shown in blue).

# F    CLOSURE AND ELIMINATION SUBSPACES

In this section, we provide additional details about results related to the closure and elimination subspaces described in Section 5.4.

At the left-slot of a given fact (e.g., $hc=$), the "goal" of the model is to predict any variable that could be associated with the left-slot variable (e.g., $h$). This requires identifying all variables previously connected to $h$ in the context. This set of variables is precisely what we call the "**closure**" of the group. We find the model is very good at this task, producing near-uniform logits over all previously seen elements of the query's group, an example of which is shown in Figure 15.

We quantify the model's ability to compute the closure by measuring the top-$K$ matching accuracy at the left-slot position. We identify the $K$ variables with the highest predicted logits. Top-$K$ matching accuracy is then computed as the proportion of these top-$K$ predictions that correspond to variables from the corresponding group $G$ that have appeared in the context so far. Perfect performance means the model assigns the $K$ highest logits exactly to the $K$ group members seen in context, regardless of their relative ordering. We also report top-1 accuracy, which is whether the highest logit is one of the variables in $G$. Over a batch of 2000 randomly sampled algebra sequences with 200 facts, we find that our model gets 100.0% top-1 accuracy, and 100.0% top-$K$ matching accuracy, indicating it has correctly learned how to compute within-group closure.

## F.1    HOW ARE CLOSURE AND CANCELLATION SETS COMPUTED?

In this section, we investigate how the set difference operation introduced in Section 5.4 is implemented by the transformer model. For a given query $xy=$, recall that the closure set $S_{\text{closure}}$ contains all elements that have previously appeared in the context associated with $x$ or $y$, and the cancellation set $S_{\text{cancel}}$ is the set of answers from previously seen facts that share either $x$ in its left-slot or $y$ in its right-slot. Upon examining attention patterns and attention head outputs via the logit lens (Nostalgebraist, 2020), we find evidence that these two sets are built up from contributions across several attention heads. We find that the closure computation is reused across token positions, and also describe a few heads implicated in constructing the cancellation set in more detail below.

**Closure Computation.** We find that the same circuitry used to compute group closure is reused across token positions: both at the left-slot prediction, and at the predictive token ('='). While we measure the model's ability to compute closure using facts' left-slot predictions, we also find that group closure dominates predictions at the predictive token during early phases of training – producing near-equal logits of group symbols until other strategies are learned (Figure 6b, orange).

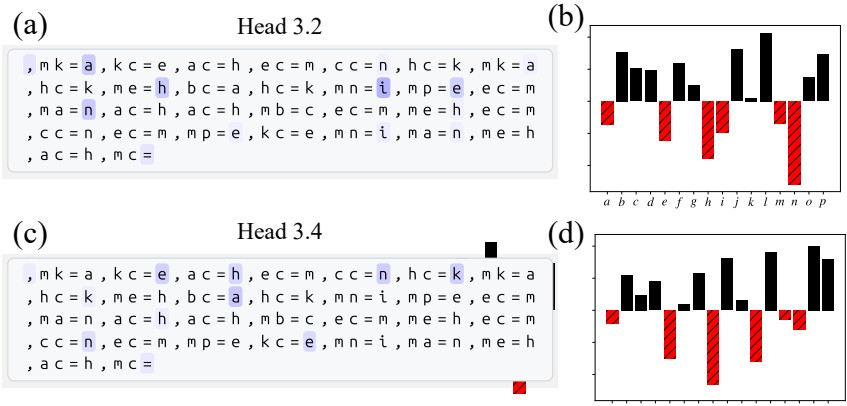

Figure 16: Cancellation Set Construction. Several attention heads in the model work together to build the cancellation set and contribute to the cancellation subspace. (a) Typical attention pattern of Head 3.2, which primarily attends to the answer-slot of facts that share the same symbol as the query's left-slot. (b) The attended-to tokens (e.g., $\{a, h, i, e, n\}$) have their logits demoted from head 3.2's output contribution (red). (c) Typical attention pattern of Head 3.4, which primarily attends to the answer-slot of facts that share the same symbol as the query's right-slot and (d) similarly demotes the attended-to token's (e.g. $\{e, h, n, k, a\}$) logits (red).

Without additional structural information (e.g., that variables represent group elements), randomly predicting a symbol from the closure is a near-optimal strategy. But as data-specific structure is learned, the closure prediction is built upon by other mechanisms such as the elimination subspace.

**Cancellation Set Construction.** A few attention heads at the final predictive token exhibit attention patterns that are suggestive of partial cancellation law behavior. For example, head 3.2 primarily attends to answer-slots of facts that share the same symbol in its left-slot as the query (i.e., facts of the form $x? = ?$, where ? can be any variable token, see Figure 16a). We find that head 3.2 places an average of $74.4\%$ of its attention probability mass on answer-slots of facts that share the same left-slot symbol as the query (averaged over 200 prompts). After attending to these tokens, head 3.2's attention contribution subsequently demotes the logits of each answer token (Figure 16b). We find another attention head (layer 3, head index 4) that primarily attends to answer-slots of facts that share the same symbol in its right-slot as the query (i.e., facts of the form $?y = ?$, see Figure 16c), doing so $57.1\%$ of the time. Similarly, head 3.4 demotes the logits of the answer-slot tokens it attends to (Figure 16d). These examples show how multiple attention heads help identify a set of tokens that should be eliminated as answers, and we find that learning a low-dimensional subspace over the attention layer can cleanly capture the corresponding cancellation subspace.

## F.2 SUBSPACE CONSTRUCTION

Here we describe how we construct a learnable subspace that can characterize multi-dimensional high-level variables such as the closure and cancellation sets described in Section 5.4.

We learn a set of Householder unit-vectors $\{v_i \in \mathbb{R}^d, ||v_i||=1\}$ (where $d$ is the model's hidden dimension), to construct a series of Householder matrices, $H_i = I - 2v_i v_i^T$, that are composed to form an orthogonal matrix $Q = H_k H_{k-1} \cdots H_1 \in \mathbb{R}^{d \times d}$, (Householder, 1958). The first $k$ columns of $Q$, denoted $Q_k \in \mathbb{R}^{d \times k}$, form an orthonormal basis for our intervention subspace. We construct our subspace projection as $W = Q_k Q_k^T$ and perform subspace-level interventions by mixing information between activations of the model on sequences $s$ and $s'$ as shown in Equation 10:

$$W h_s + (I - W) h_{s'} \to h_{s'} \tag{10}$$

where $h_s$ represents an activation taken from the model under sequence $s$, $h_{s'}$ represents an activation taken from the same location under sequence $s'$, and $\to$ means the activation $h_{s'}$ is replaced with the intervened representation $W h_s + (I - W) h_{s'}$. While we use $h_s$ to denote "activation" here, it could be a representation taken from any location in the model.

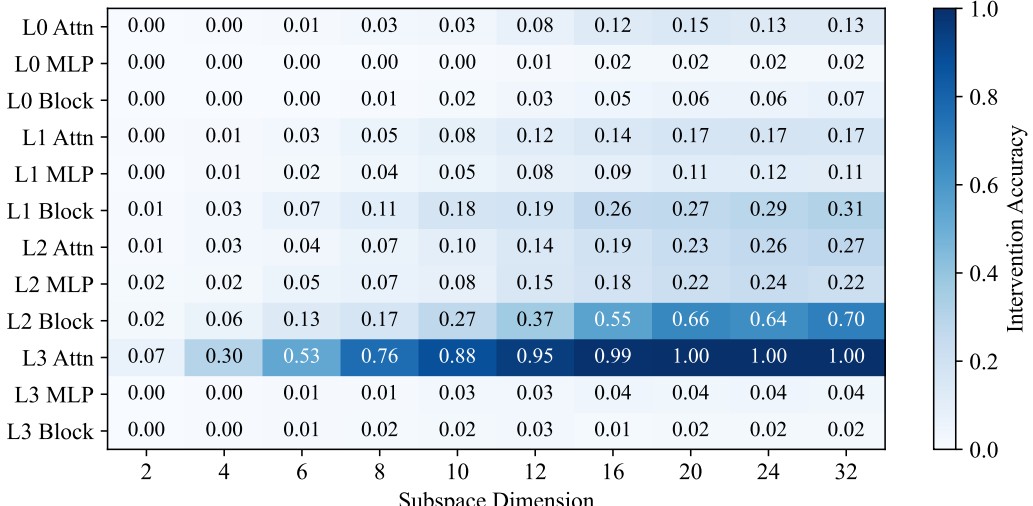

Figure 17: Identifying the closure subspace. To determine the intrinsic dimension and location of where the closure set is represented by the model we train a closure subspaces, sweeping over model components (attention layer output, MLP layer output, and full layer output) and over the number of dimensions ($k \in \{2, 4, 6, 8, 10, 12, 16, 20, 24, 32\}$). For each configuration, we report the validation accuracy of intervening with that subspace. We find that the closure set computation is best captured ($\geq 99\%$) in the final layer attention output with an intrinsic dimension of $\sim 16$, which is exactly the number of variable tokens in the model's vocabulary.

### F.2.1 SUBSPACE TRAINING DETAILS

We train closure and elimination subspaces over different model components (attention layers, MLP layers, full layers), and sweep over different values for the subspace dimension $k \in \{2, 4, 6, 8, 10, 12, 16, 20, 24, 32\}$. We measure the interchange intervention accuracy (Wu et al., 2023; Prakash et al., 2025) for each trained subspace and report the results in Figure 17.

We find that the closure set computation is best represented in the final attention layer and that a 16-dimensional subspace is sufficient to achieve near-perfect intervention accuracy (99.9%), and that it converges pretty quickly during training (Figure 18a). Subspaces trained at most locations are unable to meaningfully influence the construction of the closure set. It can be partially controlled ($\sim 70\%$) by a subspace trained on the input to the final attention layer (i.e., the layer 2 output is the same location as the layer 3 attention's input), but requires more dimensions and doesn't achieve as clean of performance as the final attention layer (Figure 17).

For the elimination subspace, we find that training on counterfactual sequence pairs similar to those used for the closure subspace ($d = 16$) results in a subspace that plateaus around $\sim 75\%$ intervention accuracy, even after extended training (Figure 18b, red). Analyzing the subspace intervention failure cases reveals that 99.9% of these sequence pairs fit into one of two categories: either (i) at least one sequence ends in an identity fact, or (ii) the counterfactual target renders the query an identity fact under intervention. Because of this, we choose to exclude sequences that end in identity facts (and counterfactually-imposed identity facts) from the training distribution of the elimination subspace. We find that without identity sequences, the elimination subspace quickly achieves near-perfect intervention accuracy (Figure 18b, blue).

The inability of the elimination subspace to capture the model's predictions on identity sequences suggests that the mechanism the model uses to solve identity sequences is distinct from the closure-based cancellation mechanism under study here. The identity mechanism is explored in more detail in Section 5.3. The fact that identity facts are easily separated from other facts under PCA might explain some of the reason why training the elimination subspace fails to converge on identity sequences, though this distinction doesn't matter for training the closure subspace (Figure 18a, red).

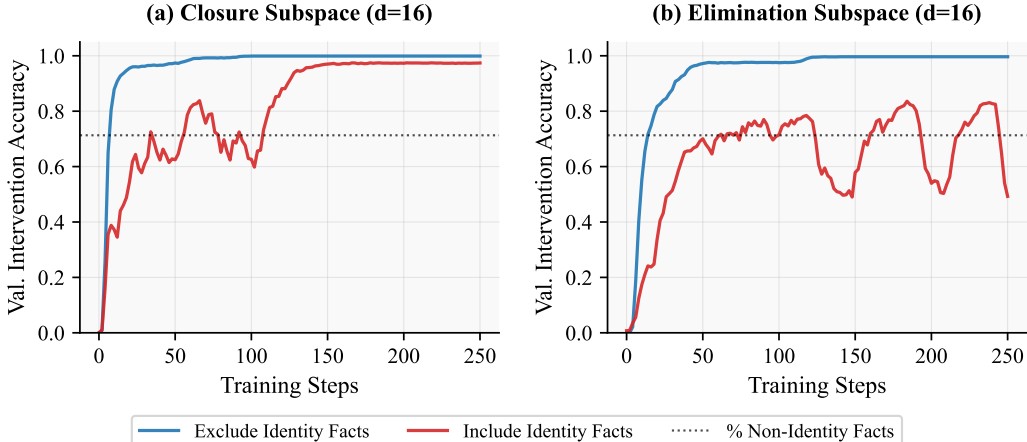

Figure 18: Subspace Intervention Accuracy. We train subspaces to represent the closure set and cancellation set used by the model (§ 5.4). (a) The closure set promotes the variables of a particular group in a sequence can be well-approximated by a 16-dimensional subspace (which is the same as the number of variables in the model's vocabulary), and achieves very good intervention accuracy both when identity facts are excluded (99.9%, blue) or included (97.5%, red) in its training data. (b) The cancellation set eliminates possible answers via the cancellation law and can also be well-approximated by a 16-dimensional elimination subspace if identity sequences are excluded from its training data (blue). When identity facts are included, the trained subspace only achieves ~75% intervention accuracy, suggesting that identity facts are solved differently. Interestingly, the percentage of the holdout data distribution that are not identity sequences is 71.1% (§ 4.1), shown as a dotted black line, and this roughly aligns with the intervention plateau of the elimination subspace.

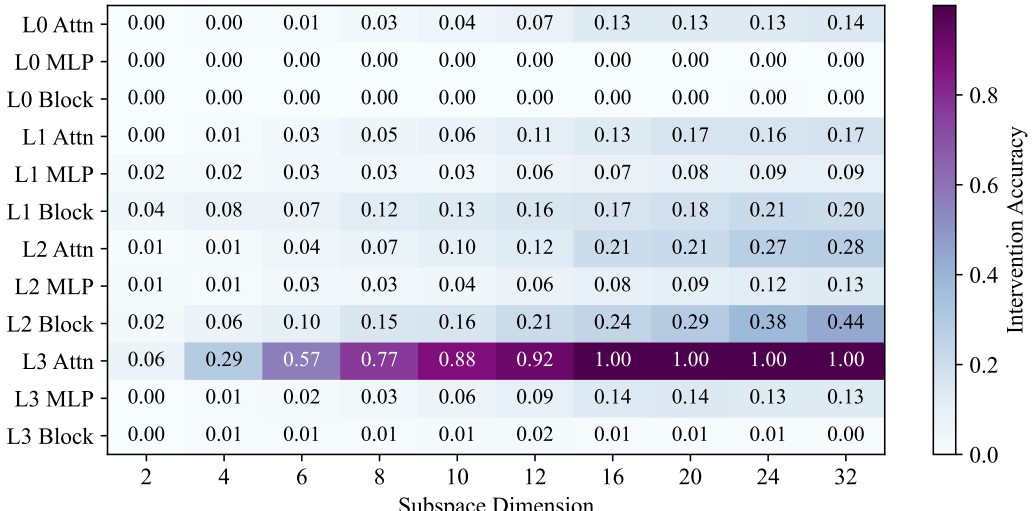

Figure 19: Identifying the elimination subspace. We also train elimination subspaces, sweeping over model components (attention layer output, MLP layer output, and full layer output) and over the number of dimensions ($k \in \{2, 4, 6, 8, 10, 12, 16, 20, 24, 32\}$) to determine where the cancellation set is represented by the model. For each configuration, we report the validation accuracy of intervening with that subspace. We find that the cancellation set computation is also best captured ($\geq 99\%$) in the final layer attention output with an intrinsic dimension of ~16, the same as the number of variable tokens in the model's vocabulary.

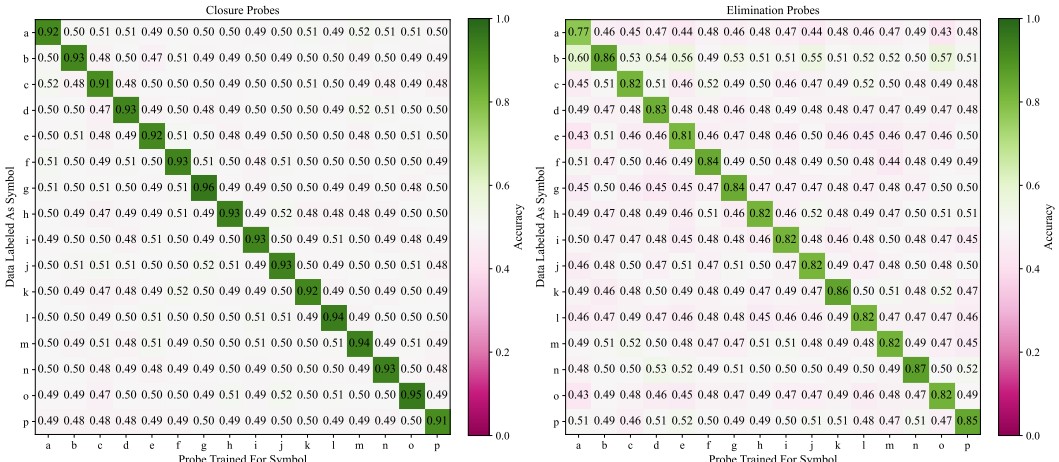

Figure 20: Closure and elimination probe generalization accuracy. We train individual probes (column) on the closure and elimination subspaces for each symbol (row) to test for the presence of each variable in the closure set or cancellation set. (left) Probe are accurately able to predict closure set membership within the closure subspace. Probes achieve between $91-96\%$ accuracy for the symbol they are trained on, and chance performance on all other symbols. (right) Probe accuracy for predicting cancellation set membership using the elimination subspace. Overall accuracy is lower than for the closure probes, but still decent overall. The probe for $a$ achieves $77\%$ accuracy, perhaps due to the fact that during training $a$ was never seen as the identity element, but all other probes classify elimination-membership with $81-87\%$ accuracy.

We also perform a sweep over layers and dimensions for the elimination subspace and find that the cancellation set is best represented in the final attention layer with a subspace of $\sim16$ dimensions (Figure 19). While it is the same size as the closure subspace, we find these subspaces are distinct from each other, as they are unable to produce valid counterfactuals in the opposite setting.

### F.2.2 PROBING THE CLOSURE AND ELIMINATION SUBSPACES

We train probes to better characterize how the trained closure and elimination subspaces represent the closure set and cancellation set. We train individual probes to detect whether a variable is in the group closure or cancellation set using both the closure and elimination subspaces (Figure 20). We find the probes can accurately predict with high accuracy whether variables are in the closure set ($91-96\%$) (Figure 20, left), and with good, but slightly worse accuracy for predicting whether a variable is in the cancellation set ($77-87\%$) (Figure 20, right).

We compare the directions of the learned probes with the unembedding directions of the model to check how aligned they are with the output space. We find that the closure probes show a weak positive alignment (average cosine similarity of $+0.38$) with the unembedding vectors with the corresponding variable (Figure 21, left). This matches our intuition that the closure subspace does in fact promote (increase the probability of) the variables that are present in the group closure.

In contrast, the elimination probes show a weak negative alignment (average cosine similarity of $-0.17$) with their corresponding unembedding vectors and near-zero similarity with off-diagonal entries (Figure 21, right). This suggests the elimination subspace "demotes", or lowers the probability of variables that are present in the cancellation set, again matching our intuition from Section 5.4.

We also compute the cosine similarity between the trained probes for both subspaces and find that for each subspace the probe for one symbol is independent of those for other symbols (Figure 22).

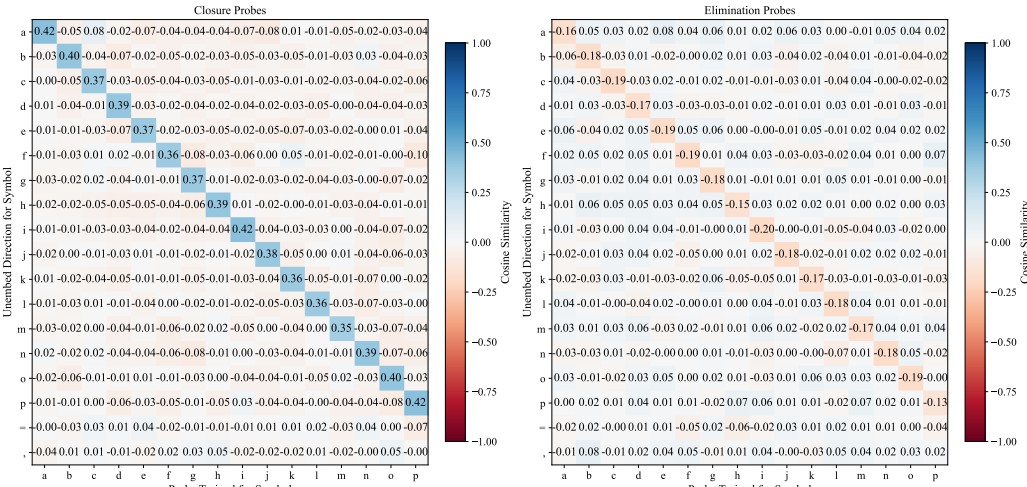

Figure 21: Cosine similarity between learned probe directions (columns) and model unembedding vectors (rows) for the closure and elimination subspaces. (left) The probe directions in the closure subspace weak positive alignment with the model's unembedding direction for their respective variable with an average cosine similarity of $+0.38$, suggesting the closure subspace does indeed promote those tokens when present. (right) The probe directions in the elimination subspace show weak negative alignment with the model's unembedding direction for their respective variables, with an average cosine similarity of $-0.17$, suggesting the elimination subspace demotes the those tokens when present, though the effect is less strong than seen for the closure subspace.

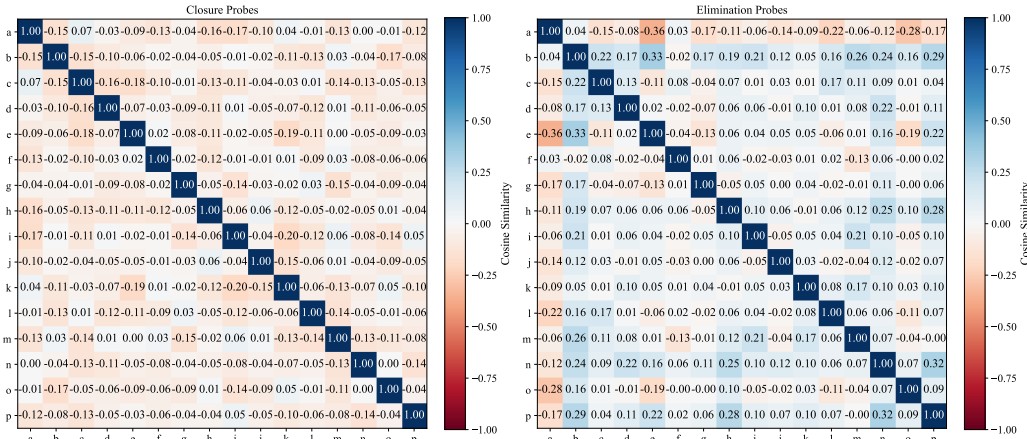

Figure 22: Cosine similarity between learned probe directions for closure (left) and elimination (right) subspaces. Each probe is trained to detect a specific symbol in the respective set. High diagonal values indicate probes learn distinct, symbol-specific directions, while off-diagonal values near zero suggest orthogonality between different probe directions. The elimination subspace probe for the $a$ variable shows a distinct trend from other probes — it has mostly negative cosine similarity with the other elimination probes — which might be explained by the fact that $a$ was not seen as an identity element during training.

# G DATA COVERAGE PSEUDOCODE

In this section, we provide some pseudo-code examples showing how we check the coverage of each algorithm hypothesized in Section 4. Optimized code implementations can be found through our project website: algebra.baulab.info.

```python
def check_copyable(sequence):
    """ sequence (str): A sequence of consecutive algebra facts.
    ex: ',fk=i,kn=g,cd=d,kh=c,in=c,nf=h,cg=g,if=n,gf=c,id=h,cg=g,df=g'
    """
    facts = sequence.split(',')
    query = facts[-1]
    return any([fact.split('=')[0] == query.split('=')[0]
                for fact in facts[:-1]])
```

Code Block 1: Verbatim Copying. A python implementation to check if verbatim copying could solve the given algebra sequence.

```python
def check_reverse_copyable(sequence):
    """ sequence (str): A sequence of consecutive algebra facts.
    ex: ',fk=i,kn=g,cd=d,kh=c,in=c,nf=h,cg=g,if=n,gf=c,id=h,cg=g,df=g'
    """
    facts = sequence.split(',')
    query = facts[-1]
    return any([fact.split('=')[0] == query.split('=')[0][::-1]
                for fact in facts[:-1]])
```

Code Block 2: Commutative Copying. Python implementation to check if commutative copying could solve the given algebra sequence.

```python
def check_identity_solvable(sequence):
    """ sequence (str): A sequence of consecutive algebra facts.
    ex: ',fk=i,kn=g,cd=d,kh=c,in=c,nf=h,cg=g,if=n,gf=c,id=h,cg=g,df=g'
    """
    facts = sequence.split(',')
    query = facts[-1]

    left_identity = [fact[0] == fact[-1] and fact[1] in query.split('=')[0] for fact in
     facts[1:-1]]
    right_identity = [fact[1] == fact[-1] and fact[0] in query.split('=')[0] for fact in
     facts[1:-1]]

    return any(left_identity or right_identity)
```

Code Block 3: Identity Recognition. Python implementation to check if identity recognition could solve the given algebra sequence.

```python
def check_closure_elimination_solvable(sequence):
    """ sequence (str): A sequence of consecutive algebra facts.
    ex: ',fk=i,kn=g,cd=d,kh=c,in=c,nf=h,cg=g,if=n,gf=c,id=h,cg=g,df=g'
    """
    facts = sequence.split(',')
    query = facts[-1]

    share_symbol = [fact for fact in facts[1:-1] if query[0] in fact or query[1] in fact]

    share_a_on_left = [fact for fact in facts[1:-1] if fact[0] == query[0]]
    share_b_on_right = [fact for fact in facts[1:-1] if fact[1] == query[1]]

    share_symbol_slots = share_a_on_left + share_b_on_right

    def get_closure_set(facts):
        return set(''.join([x for x in facts]).replace('=', ''))

    set_closure = get_closure_set(share_symbol) # includes answers
    answer_closure = get_closure_set([x[-1] for x in share_symbol_slots])

    return len(set_closure - answer_closure) == 1 and (set_closure - answer_closure) ==
     sequence[-1]
```

Code Block 4: Closure-based Cancellation. Python implementation to check if a closure-based elimination rule could solve the given algebra sequence.

```python
def check_associative(sequence):
    """
    sequence (str): A sequence of consecutive algebra facts.
    ex: ',fk=i,kn=g,cd=d,kh=c,in=c,nf=h,cg=g,if=n,gf=c,id=h,cg=g,df=g'
    """
    facts = sequence.split(',')
    query = facts[-1]

    triplets = determine_associative_pairs(query)

    is_associative=False
    for triplet in triplets:
        all_facts_exist = True
        for fact in triplet:
            if fact not in facts: # Need each fact of an associative triplet
                all_facts_exist=False
                break
        if all_facts_exist: # If there's a triplet of facts that compose to solve the query
            is_associative=True
            break

    return is_associative
```

Code Block 5: Associativity. Python implementation to check if composition of facts via associativity could solve the given algebra sequence.

## H  USE OF LARGE LANGUAGE MODELS

As per the ICLR 2026 author guidelines, we provide details about our use of large language models (LLMs) in the preparation of this manuscript. LLMs were primarily used as a general-purpose tool to aid and polish writing, both at the sentence level (e.g., grammar or re-wording sentences), and at the paragraph level (e.g., re-organizing sentences in a paragraph). When considering LLM suggestions, the resulting text went through many subsequent editing rounds. LLM use did not contribute in any way that we would consider equal to the level of an author.

