# OpenReview forum: "In-Context Algebra"
_ICLR.cc/2026/Conference — ICLR 2026 Poster_

### Official Review · Reviewer_9LQ5 · 2025-10-26

**Soundness:** 3
**Presentation:** 3
**Contribution:** 2
**Rating:** 2
**Confidence:** 4

**Summary:**

The paper investigates the mechanisms transformers learn to solve arithmetic problems defined by finite algebraic groups. Critically, the task is set in an in-context learning format where the meaning of tokens (their assignment to specific group elements) is variable and defined only within the current sequence. This contrasts with prior work (e.g., on "grokking" modular arithmetic) which used fixed token-to-value mappings and found the emergence of geometric, Fourier-based representations.

The authors find that in their variable-token setting, the model achieves high accuracy and generalizes to unseen groups. Through causal analysis (activation patching, subspace analysis), they identify a different set of mechanisms: not geometric embeddings, but "symbolic" operations. These include a dedicated head for verbatim and commutative copying, a two-part mechanism for identity element recognition (query promotion + identity demotion), and a subspace-based mechanism for closure-based cancellation (tracking group membership and eliminating invalid answers). The paper concludes that geometric representations may be an artifact of fixed-symbol tasks and that models default to symbolic reasoning when meanings are context-dependent.

**Strengths:**

- The "in-context algebra" task is a novel and interesting probe for mechanistic interpretability. By forcing token meanings to be sequence-local, it creates a clean setting to study how transformers perform reasoning when they cannot rely on pre-trained, fixed-meaning embeddings.

- The paper goes beyond simple performance metrics and attempts a rigorous mechanistic decomposition of the learned algorithm. The use of causal interventions (patching) to isolate specific heads (e.g., Head 3.6 for copying) and the PCA/subspace analysis to understand identity and closure mechanisms are commendable.

- The finding that mechanisms are multi-part (e.g., identity recognition being a combination of "query promotion" and "identity demotion") is insightful and adds a layer of depth beyond just "this head does X."

**Weaknesses:**

- The paper's central thesis is that its contribution is "challenge the hypothesis that geometric embeddings are the primary mechanism" for algebraic problems. This framing may be slightly marginal for a conference. The prior work on geometric/Fourier-basis representations documented the phenomenon specific to a task with fixed token meanings. It might be more constructive to frame this paper's findings as a complement to that work, showing an expected and important contrast: when the pre-conditions for geometric embeddings are removed (i.e., by making token meanings variable), models develop a different, symbolic-like strategy. This distinction is a valuable contribution in itself, showing how task design dictates the learned mechanisms.

- *Weak Experimental Analysis*: The paper's mechanistic analysis is a great start, but feels incomplete in a few areas that would make the claims more robust.
  - **Omission of Associativity**: The authors hypothesize five mechanisms (Sec 4) but find their model performs poorly on "Associative Composition" (60.2% accuracy at k=50, Fig 3c). Associativity is arguably the most fundamental property of a group. The paper would be much stronger if it included a causal analysis of this failure. Why does the model fail at associativity? What partial or heuristic-based mechanism does it learn instead? Analyzing this failure mode could offer even deeper insights than analyzing the successes.
  - Because the analysis (understandably) focuses on the learned mechanisms, it's less clear what generalizable insights this task provides. The mechanisms (copying, identity, elimination) are powerful heuristics, but they aren't unique to group theory. It would be great if the authors could further discuss whether the model has learned "algebra" or a set of clever, task-specific heuristics.

- *Ablation Studies*: The analysis is based on a single "toy" model (4-layer, 8-head) trained from scratch. The findings would be more generalizable if the authors could provide some ablation studies. For example, how do these mechanisms change with model depth/width, or other hyperparameters? This would help confirm that these mechanisms (e.g., Head 3.6) are a general solution rather than an artifact of one specific architecture.

- "From Scratch" vs. Pre-trained: Relatedly, the "from scratch" setting is interesting, but a very relevant follow-up would be to explore how a pre-trained LLM tackles this task. Would it develop these same mechanisms, or would it repurpose existing in-context learning circuits? Exploring this could bridge the gap between toy models and real-world LLMs.

- **Tokenization Details**: A small but important detail that would be good to clarify is tokenization. The paper implies each variable is a single token. It would be helpful to explicitly confirm this and perhaps discuss how the task difficulty might change if variables were multi-token strings, which is a common scenario in natural language.

- I also notice authors may also may be great mention some relevant work like [Language Models are Symbolic Learners in Arithmetic](https://arxiv.org/abs/2410.15580) which they use subgroups to derive symbolic behaviors in LM and a concurrent work [Why Can't Transformers Learn Multiplication? Reverse-Engineering Reveals Long-Range Dependency Pitfalls](https://arxiv.org/abs/2510.00184) (concurrent). Situating this paper's findings relative to these would be very interesting for the community.

**Questions:**

- The model's difficulty with "Associative Composition" (Fig 3c) is very interesting, as this is a core algebraic property. I was surprised to see it wasn't included in the causal analysis. Could the authors elaborate on why this was omitted, and perhaps share any preliminary findings on why the model struggles here? This seems like a fascinating area for future work.
- Could the authors provide more mechanistic detail on the "Closure-Based Cancellation" (Sec 5.4)? The text describes it as a set difference $S_{closure}$ - $S_{cancel}$. How are these sets represented in their respective subspaces, and how is the "set difference" operation mechanistically computed by the model? The high-level concept is clear, but a deeper dive into the "how" would be fantastic.
- The model's ability to generalize to unseen groups (Fig 2c) while failing at associativity is a curious result. Does this suggest to the authors that the model has learned general heuristics (copying, identity, cancellation) that work for many group-like structures, rather than a deep, rule-based understanding of group axioms? The generalization to non-groups (semigroups, magmas) seems to support this, and I'd be interested in the authors' take.

---

> ### Author Response · Authors · 2025-11-21
> **Response to Reviewer 9LQ5 (1/2)**
>
> Thank you for your detailed review and helpful feedback! We are glad you found our work novel and our findings about the identity mechanism insightful. We have updated our paper submission to answer your questions and incorporate your feedback. We also address your questions here:
>
> > **Central Framing: Variable Setting as Complementary to Prior Work**
> - Thank you for raising this point. We agree that our contributions are both distinct from and complementary to prior work studying fourier representations, since we study a setting where tokens have no fixed meaning. To better reflect this framing, we have made several changes throughout the paper, mainly concentrated in the conclusion section. The conclusion now reads as follows: “We have studied LMs trained on a focused algebra task designed to isolate abstract in-context reasoning behavior in the absence of fixed-meaning symbols. Our findings suggest that the kinds of reasoning strategies learned by transformers are dependent on the task structure. In our in-context algebra setting, where tokens carry no fixed meaning, we have analyzed the mechanisms learned by transformer LMs in detail and found that models develop symbolic mechanisms instead of the familiar geometric strategies found in settings where tokens do have fixed meanings. We have seen that transformers can learn to manipulate symbols in-context without needing to refer to their underlying meaning, similar to the way that high-school algebra students learn to solve math problems by manipulating letter variables without constantly thinking about the values they might contain. Understanding when and why models choose different computational strategies remains an important open question for future interpretability work.”
>
> > **Ablation Studies:**
> - Thank you for the suggestion. We trained and analyzed 16 additional models sweeping over layers, heads, dimensions, and data mixing. The results of these ablations are detailed in Appendix B. We find that for almost all configurations that we train, models consistently learn similar mechanisms in the same order. However, models trained with too small of a hidden dimension did not have enough capacity to learn the task. We also found that across all configurations, models struggled to solve associativity sequences, with no model getting above 90%.
> - The high-level mechanisms learned by models trained with the same hyperparameters, but different seeds were also consistent. While the particular head index was not the same for heads performing copying or query promotion, we did still see this same functionality appear across seeds, suggesting these mechanisms are generalizable for the in-context algebra task.
>
> > **In-Context Algebra for LLMs**
> - We agree that it would be interesting to investigate this setting on pretrained LLMs, though it would be an extensive undertaking. Perhaps the most related work to this suggestion is  [Yang et al] who study algebraic rule induction in large pretrained transformers.
> - While there is likely not enough time during the rebuttal process to fully mechanistically understand a large pretrained LLM on the in-context algebra task, there may be adequate time to benchmark an LLM on this task and for initial investigations into common/shared attention patterns between the small transformer setting and the LLM setting on different data subsets. If the reviewer is interested, we can provide an update on preliminary findings in this direction in a future discussion comment.
>
> > **Tokenization Details:** :How might the task difficulty change if variables were multitoken strings?"
>
> - You are correct. In our setting, each variable is a single token. There are 16 variable tokens, a predictive token ‘=’, and a separator token ‘,’ for a total of 18 tokens. We have made this more explicit in Line 86 of the paper, by adding the phrase “length-one”. (i.e. “Training data consists of sequences of length-one variable tokens $v_i \in V$...”)
> - Introducing variables that are multitoken strings into this setting would certainly make the problem harder - the model would likely first have to learn how to “detokenize” subwords, and which tokens can be combined together into abstract variables before being able to implement symbolic algorithms on top of them. This would also likely require more layers to do than our current task formulation. There are several papers presenting evidence that pretrained LMs naturally implement a  “detokenization” process in early layers where abstract concepts are formed by combining individual subword tokens [Nanda et al, Feucht et al, Kaplan et al].

---

> > ### Author Response · Authors · 2025-11-21
> > **Response to Reviewer 9LQ5 (2/2)**
> >
> > > **Related Work Comparison**
> >  - Thank you for pointing out these additional related works, we have incorporated them into our related work section. To summarize here, the findings of [Deng et al] and [Bai et al] complement each other nicely. [Deng et al] studies LLMs fine-tuned for arithmetic and finds that they do not use partial products, but instead rely on symbolic “subgroup matching”. In contrast, [Bai et al] show that when transformers are fine-tuned with implicit chain of thought, they do learn to leverage partial products for multi-digit arithmetic and represent their numbers in a fourier basis. These works both study arithmetic in settings where tokens have fixed meaning, but this is an interesting case of two qualitatively different behaviors exhibited in LMs: symbolic reasoning vs. geometric reasoning. Our work shows that taking a familiar arithmetic setting and removing token meanings causes models to develop different, symbolic algorithms, in contrast to geometric algorithms seen in the fixed-token meaning setting. This is similar to the contrast between the results of [Deng et al] and [Bai et al].
> >
> > > "**More Details on Associativity Analysis:** The paper would be much stronger if it included a causal analysis of this failure. Why does the model fail at associativity? What partial or heuristic-based mechanism does it learn instead? Could the Authors elaborate on why this was omitted, and share any preliminary findings on why the model struggles here?"
> >
> > - Thank you for your interest in our problem setting, and around associativity in particular. We also find it fascinating! We agree that understanding how/whether the model implements some form of associative composition would strengthen the paper. We have spent some time trying to understand how the model solves associative sequences, but have not made much progress yet since the model performance is much lower than it was for other “crisper” mechanisms.
> > - We will update this thread as more details are found, that either help characterize failure cases or partial successes.
> > - Part of the reason we excluded it initially was because we do not yet have a detailed understanding of how the model solves associative sequences (even if only partially).
> >
> > > **More Details on Closure-based Cancellation**
> > - We provide more detail on the implementation of the closure-based cancellation mechanism in Appendix D. We find that the two sets are formed through the coordination of several attention heads in the final attention layer.
> >
> >
> > ___
> > Nanda et al [Fact Finding: Attempting to Reverse Engineer Factual Recall on the Neuron Level](https://www.alignmentforum.org/posts/iGuwZTHWb6DFY3sKB/fact-finding-attempting-to-reverse-engineer-factual-recall)
> >
> > Feucht et al [Token Erasure as a Footprint of Implicit Vocabulary in LLMs](https://aclanthology.org/2024.emnlp-main.543/)
> >
> > Kaplan et al [From Tokens to Words: On the Inner Lexicon of LLMs](https://openreview.net/forum?id=328vch6tRs)
> >
> > Yang et al [Emergent Symbolic Mechanisms Support Abstract Reasoning in Large Language Models](https://openreview.net/forum?id=y1SnRPDWx4)
> >
> > Deng et al [Language Models are Symbolic Learnings in Arithmetic](https://arxiv.org/abs/2410.15580)
> >
> > Bai et al [Why Can’t Transformers Learn Multiplication? Reverse-Engineering Reveals Long-Range Dependency Pitfalls](https://arxiv.org/abs/2510.00184)

---

> > > ### Comment · Reviewer_9LQ5 · 2025-11-24
> > > **Response to Authors**
> > >
> > > Thank you for the response! I think the authors did a great job addressing my reviews and resolving many of my concerns. It’s clear they spent significant time and effort thinking through this problem. I still feel the contribution is slightly on the weaker side, but that’s inherently subjective. I’ll increase my score slightly to reflect their thorough revisions.

---

### Official Review · Reviewer_sQQh · 2025-10-31

**Soundness:** 4
**Presentation:** 3
**Contribution:** 3
**Rating:** 8
**Confidence:** 4

**Summary:**

Overall, this work is thorough, well-written, well-motivated, and is strongly situated in the literature. The causal experiments broadly support the claims made in the paper. While the scope of transformer models is limited (I would like to see how and if model depth changes the ability of the model to make completions of different type), the paper offers a deep dive into how transformers accomplish in-context reasoning in this scenario. This paper would be useful and interesting to those in the mechanistic interpretability field, and it should be accepted to ICLR.


The authors design a novel experiment in which small transformer models learn to complete algebraic facts where the meaning of individual symbols is new (and only defined by the context) each time. This is significant because it helps researchers understand mechanisms by which transformers learn from context rather than from pre-determined world knowledge or structure when completing tasks. The authors report that small transformers can achieve near-perfect accuracy on the task and generalize to new samples. Most of the accuracy can be attributed to five known strategies for the completions, but a small percentage of performance remains unexplained. They find three causal mechanisms by which transformers complete the task: identity element recognition, commutative copying, and closure-based cancellation. The causal experiments involve measuring IE when patching specific attention layers from completions where the strategy is possible to completions where the strategy is not possible. They find specific attentions are responsible for implementing specific strategies. These attention heads tend to deactivate (attend to themselves) when the strategy is not possible. The authors describe a clear multi-step method by which identity facts are recognized and promoted to form the completion. Lastly, the authors investigate mechanisms for closure-based cancellation, leveraging probes to track the model’s knowledge of candidate completion elements given incomplete information and subspace models which generate counterfactual answers. The mechanisms are verified with causal experiments.


The work is well-motivated and grounded in relevant literature on mechanistic interpretability and LLM arithmetic problem solving.

Minor Comments:
Line 409 “can encoding” -> “can encode”
Line 481 typo "transformner" -> "transformer"

**Strengths:**

The work is well-motivated and grounded in relevant literature on mechanistic interpretability and LLM arithmetic problem solving.

The claims regarding learning learning mechanisms are largely validated using causal experiments.

The experiment design is thorough and sound. There is sufficient reproducibility information.

The paper is well-written and easy to follow for those in the field.

**Weaknesses:**

The scope of transformer model is limited to one architecture and size. It would be interesting to see how the results change as different numbers of layers (but also attention heads, representation dimension) are considered. One might predict that transformer layer depth allows for more complex sequential operations. For example, depth might lead to better performance on associative composition, which did not appear to benefit from more in-context examples.


There is limited to no discussion of ways in which the results *might* extend beyond this abstract problem setting

**Questions:**

How does transformer size impact the mechanisms learned? How does size (mostly depth) impact the performance on this task?

---

> ### Author Response · Authors · 2025-11-21
> **Response to Reviewer sQQh**
>
> Thank you for the positive feedback, we are glad you found our paper well-written, well-motivated, and thorough!
>
> > "There is limited to no discussion of ways in which the results might extend beyond this abstract problem setting."
> - Thanks for pointing this out, while it is not contained in a single section of the paper, portions of this discussion are spread throughout the paper. We summarize it here to make it more concrete (copied from the global response):
> - Our in-context algebra setting is inspired by the fact that in natural language, much of the meaning of every word is determined by the context. However, this is entangled with the prior meaning of words that is learned from training and baked into the token embeddings of LMs. Because of this, it can be difficult to discern the impact of parametric knowledge in LMs and distinguish it from the effect of context. We designed the in-context algebra setting to cleanly isolate in-context reasoning from pre-encoded embedding information.
> - In our setting, removing the ability for tokens to have fixed value meanings showed us that transformer models can reason by manipulating references in-context, without knowing their underlying meaning. This is similar to how high-school algebra students learn to solve math problems by manipulating letter variables rather than constantly thinking about the values they might represent. This suggests it is possible for LMs to use similar symbolic strategies to solve other problems. Previous studies of the choice between “symbol-based” and “value-based” reasoning in LMs have yielded inconsistent results [Calais et al, Cheng et al], though we have seen studies of LMs consistently learning another form of referential reasoning mechanisms (e.g., binding/ordering IDs [Dai et al, Feng et al, Prakash et al.]). The mechanisms we characterize in our work seem to be task-specific, suggesting that there are potentially more referential reasoning mechanisms LMs develop that we are not yet aware of.
> - The mechanisms we identify here are largely task-specific, but seem to be algorithmically general. Regardless of the domain, if the input has a similar form to the data used to initially discover a mechanism, it’s possible to expect a similar symbolic mechanism at play. For example, copying is a symbolic mechanism whose valid “domain”  is any sequence with at least one repetition in-context. While copying has been heavily studied [Olsson et al], it would be interesting to characterize the “domain of invariance” of other symbolic mechanisms.
>
> > "It would be interesting to see how the results change as different [hyperparameters] are considered. How does transformer size impact the mechanisms learned? How does size (mostly depth) impact the performance on this task?"
>
> - We agree, thank you for the suggestion! We have added several new results to Appendix B that examine various hyperparameter ablations (B.1, B.2). We trained 16 new models with different hyperparameter configurations to study the effect of 4 hyperparameters: (1) number of layers, (2) model hidden dimension, (3) number of attention heads per layer, and (4) group mixing probability.
> - Across all experiments, we found that models trained on the in-context algebra task setup struggle to learn associativity. While several configurations do better than the original model we evaluate in the main paper, none of them achieve above 90% accuracy on that data subset.
> - The number of layers (depth) didn’t seem to have too much of an impact on performance. The 2-layer model achieved slightly lower evaluation accuracy (93.5%), while all models with 4 or more layers achieve 98.6% evaluation accuracy or above.
> For hidden dimension, we do see that models with hidden dimension 128 and 256 fail to achieve evaluation accuracy above 90% suggesting low capacity models can’t solve the task in the allotted training steps. We see that higher capacity models typically lead to better performance and quicker convergence to high evaluation accuracy.
>
> > Minor Typos:
> - We appreciate you pointing these out, they have been updated.
>
> ___
> For links to citations, see the response to all reviewers.

---

### Official Review · Reviewer_XJFb · 2025-11-01

**Soundness:** 3
**Presentation:** 3
**Contribution:** 2
**Rating:** 6
**Confidence:** 4

**Summary:**

This work explores how transformers learn to perform certain types of symbolic manipulations, where tokens have no fixed meaning but rather have meaning that is context-dependent. This is done through an in-context learning algebra task where the model is prompted with a sequence of $k$ facts from a set of groups whose elements are randomly mapped to tokens in the vocabulary, then asked about some product in one of the groups. This design makes it so that the learned embeddings for each token cannot themselves represent the identities of the group elements. Instead, the model must learn the corresponding algebra in-context. The authors identify five simple strategies (verbatim copying, commutative copying, identity element recognition, associative composition, and closure-based cancellation) that account for some of the model's performance. They then identify how the model implements four out of five of these mechanisms in causal intervention techniques.

**Strengths:**

- The problem is interesting and well-motivated. The paper is generally well-written and clearly presented.
- The methodology is sound, and the conclusions are moderate and well-supported by the empirical evidence.
- The in-context algebra task provides a reasonable testbed for studying symbolic reasoning, which ablates the effect of learned geometric embeddings.
- The analysis of the closure-based cancellation mechanism is technically solid and one of the paper’s strongest contributions.

**Weaknesses:**

- The task probes a narrow type of symbolic reasoning. The five hypothesized strategies are relatively simple and do not fully capture the richer “meaning-free” symbolic manipulation motivating the study. The contribution would be stronger if the task enabled the discovery of more complex or novel mechanisms.
- The causal analysis fails to identify a concrete mechanism for associative composition, arguably the most conceptually interesting of the five. The mechanisms that are identified for the other strategies are similar to transformer circuits that were identified in previous work.
- The five hypothesized mechanisms do not fully explain model performance, especially for contexts with fewer facts $k$ (which is the more challenging and more interesting case)

See also the questions below.

**Questions:**

- Is the task designed to be solvable for all numbers of facts $k$? I.e., does there exist a unique solution for each prompt sequence? From the random nature of the data generation process described, it seemed like the answer is no. If so, it would be interesting to compare the model performance to the ideal algorithmic performance. How close does the transformer model get to this optimal level of performance?
- Why is the "associative composition" strategy missing from Figure 3?
- What was the purpose of having the task be composed of multiple groups? I can imagine an alternative (simpler) task where, in each sequence, a group structure is randomly generated with group elements being randomly assigned tokens in the vocabulary. What would the difference be between these two tasks for the purposes of the investigation?

Minor comments:
- the bold "=" sign does not render very nicely (the spacing is a bit off) in Eqs (2) and (3). It makes the two equations appear a bit out of alignment.
- In Fig. 2a, you can use different colors rather than line styles for different lines to improve readability.
- The "Task Description" section was a bit unclear on the first read. I'd suggest revising it. For example, the sentence in lines 95-96 was phrased in a confusing way (needed to look at the figure to understand). I'd also recommend explicitly mentioning that the mapping $\phi_s$ is one-to-one and that $|H_s| < |V|$

---

> ### Author Response · Authors · 2025-11-21
> **Response to Reviewer XJFb (1/2)**
>
> Thank you for your review, we’re glad you found the problem setting interesting, the paper clearly presented, and our conclusions well-supported by empirical evidence.
>
> > “The causal analysis fails to identify a concrete mechanism for associative composition, arguably the most conceptually interesting of the five. The mechanisms that are identified for the other strategies are similar to transformer circuits that were identified in previous work.”
>
> - We agree that it would be interesting to understand how/whether the model implements some form of associative composition. We have spent some time trying to understand how the model solves associative sequences, but have not made much progress yet, in part because the model performance is much lower than it was for other “crisper” mechanisms. However, the other mechanisms we characterize in the paper do account for a large portion of the overall training and test data.
> - We will update this thread as more details are found about associativity, that either help characterize failure cases or partial successes.
> - We agree that copying (and commutative copying) are mechanisms that have been often studied in prior work. However, to the best of our knowledge, we have not seen *set-oriented* mechanisms such as the closure-based cancellation mechanism described in prior work. Similarly, the combination of a query promotion submechanism and identity demotion submechanism that work together to solve identity facts was an intuitive, yet clever implementation we had not seen described in prior work. If there is specific work that these mechanisms reminded you of that we may have missed, we are happy to include citations to them in a future update of the paper.
>
> > “The five hypothesized mechanisms do not fully explain model performance, especially for contexts with fewer facts  (which is the more challenging and more interesting case)”
>
> - We agree that this is a current limitation on our understanding of the problem. However, while the five mechanisms we propose do not fully explain the data distribution, they do come very close (within 3% of both training and hold-out data). After factoring in the algorithmic coverage of associativity (which we recently added to Figure 3), our 5 algorithms combined can solve 90.4% of all training sequences of up to 200 facts and 84.7% of all held-out sequences. The model achieves a corresponding AUC of 92.4% on training data and 87.3% on held-out data. The gap for each of these is only 2% and 2.6%, most prominent for contexts that have between 10 and 50 facts.
> - While we do not yet have a mechanistic account of how the model might solve associative sequences, we have made significant progress in understanding a large fraction of how the in-context algebra setting is solved.
>
> > “Is the task designed to be solvable for all numbers of facts ? I.e., does there exist a unique solution for each prompt sequence? From the random nature of the data generation process described, it seemed like the answer is no. If so, it would be interesting to compare the model performance to the ideal algorithmic performance. How close does the transformer model get to this optimal level of performance?”
> - The in-context algebra task is not guaranteed to be solvable for all numbers of facts, nor each prompt sequence. Because we allow repetitions and a mixture of groups, it is fairly easy to construct a valid (usually short) in-context algebra sequence where the answer is ambiguous or without enough information and thus does not have a unique solution.
> - It is unclear to us at the moment how to determine the “ideal algorithmic performance” for this setting. We have tried to quantify this in Section 4 with a mix of algebraic properties (i.e., five hypothesized mechanisms), but these only account for approximately 90% of all training data, and 84.7% of all held-out data.

---

> > ### Author Response · Authors · 2025-11-21
> > **Response to Reviewer XJFb (2/2)**
> >
> > > “Why is the associative composition strategy missing from Figure 3?”
> > - Part of the reason we excluded it was because we do not yet have a detailed understanding of how the model solves associative sequences (even if only partially). However, we have now added it to Figure 3, shown in blue to demonstrate its algorithmic coverage. It solves an additional 3.6% of the training data, and 16.9% of the held-out data, closing the gap to hold-out model performance to only 2.6%.
> >
> >
> > > “What was the purpose of having the task be composed of multiple groups? I can imagine an alternative (simpler) task where, in each sequence, a group structure is randomly generated with group elements being randomly assigned tokens in the vocabulary. What would the difference be between these two tasks for the purposes of the investigation?”
> >
> > - Thank you for the insightful question. We have included some new results related to this idea in Appendix B.2, where we study the effect that the group mixing probability has on training loss and evaluation performance. The mixing probability is a hyperparameter we set that determines how likely it is for additional groups to be sampled when generating a sequence. It is directly related to in-context task diversity. There are two main reasons we chose to have sequences be composed of multiple groups rather than using a single group:
> > - The first is that having multiple groups in a sequence reduces the chances that a sequence (or subsequence) can be solved via copying as the sequence length grows. This is desirable for us as we are interested in studying a model that learns mechanisms beyond simple copying. Increasing the number of groups per sequence makes this more likely.
> > - The second reason is that increased task diversity has been shown to lead to better generalization behavior [Raventos et al], [Park et al], and we wanted our model to be as good as possible. As a secondary, but somewhat counterintuitive benefit to this, we also found that the training also converged faster compared to the setting where we only select a single group per sequence (see Appendix B.2 for more details).
> >
> > > Minor Comments:
> > - Thank you for pointing out these typos and colors versus line styles, they have been updated at your suggestion.
> > - Parts of the task description in Section 2 were updated to be more explicit: specifying that $\varphi_s$ is one-to-one, and that $|H_s| \leq |V| = N$.
> > ___
> > Raventos et al [Pretraining task diversity and the emergence of non-bayesian in-context learning for regression](https://openreview.net/forum?id=BtAz4a5xDg)
> >
> > Park et al [Competition dynamics shape algorithmic phases of in-context learning](https://openreview.net/forum?id=XgH1wfHSX8)

---

> ### Author Response · Authors · 2025-11-24
> **Additional Response to Reviewer XJFb**
>
> > “The task probes a narrow type of symbolic reasoning. The five hypothesized strategies are relatively simple and do not fully capture the richer meaning-free symbolic manipulation motivating the study. The contribution would be stronger if the task enabled the discovery of more complex or novel mechanisms.”
>
> - We believe the core motivation of the work is precisely to isolate the minimal, meaning-free symbolic algorithms that transformers reliably discover when all pretrained token embeddings are removed. These mechanisms are elementary because the variable-symbol setting strips away any incentive for the model to learn richer geometric or embedding-based representations that are common in fixed-symbol algebraic tasks (e.g., Fourier-like representations or geometric embeddings in prior work). The fact that very simple mechanisms suffice to solve a large percentage of instances (and emerge consistently across random seeds and architectures) is, in our view, a surprising and central finding. That said, we agree that exploring settings where more sophisticated mechanisms are forced to emerge is an exciting direction for future work.

---

> > ### Comment · Reviewer_XJFb · 2025-11-26
> >
> > Thanks to the authors for their principled response. I appreciate their acknowledgement of limitations and their explanation of the challenges.
> >
> > *W1: narrowness in scope of "symbolic manipulation"*
> > > We believe the core motivation of the work is precisely to isolate the minimal, meaning-free symbolic algorithms that transformers reliably discover when all pretrained token embeddings are removed. These mechanisms are elementary because the variable-symbol setting strips away any incentive for the model to learn richer geometric or embedding-based representations that are common in fixed-symbol algebraic tasks (e.g., Fourier-like representations or geometric embeddings in prior work).
> >
> > My point is that the task you explore probes a narrow and specific sense of "symbolic manipulation." The mechanisms are simple because the task is simple and solvable by these simple mechanisms. Indeed, some of the mechanisms you identify are "heuristics" that work for the data distribution rather than reflecting true learning of symbolic computation.
> >
> > *W2: The causal analysis fails to identify a concrete mechanism for associative composition*
> > > We have spent some time trying to understand how the model solves associative sequences, but have not made much progress yet, in part because the model performance is much lower than it was for other “crisper” mechanisms.
> >
> > This is understandable. I think it is important to state this limitation clearly in the paper and highlight it as a direction of future work.
> >
> > *W3: The causal analysis fails to identify a concrete mechanism for associative composition, arguably the most conceptually interesting of the five. The mechanisms that are identified for the other strategies are similar to transformer circuits that were identified in previous work.*
> >
> > > We agree that this is a current limitation on our understanding of the problem. However, while the five mechanisms we propose do not fully explain the data distribution, they do come very close (within 3% of both training and hold-out data).
> >
> > Thank you for acknowledging the limitation. Again, I think it is important to highlight this limitation in the paper.
> >
> > While I take your point that the five mechanisms come close to explaining the data distribution (within 3%), this is not fully satisfactory because there may be interesting mechanisms that have been missed, and the level of coverage (3%) depends on the data generation process (i.e., you could construct a test distribution that oversamples a certain subset of inputs for which the coverage is potentially much smaller).
> >
> > *Including the associative composition strategy in Figures*
> > > Part of the reason we excluded it was because we do not yet have a detailed understanding of how the model solves associative sequences (even if only partially). However, we have now added it to Figure 3, shown in blue to demonstrate its algorithmic coverage
> >
> > Thank you. It is important to include it to provide a more complete picture, and to avoid omitting portions of the results simply because they highlight a limitation in the analysis.
> >
> >
> > ---
> >
> > Since my main concerns have not been addressed, I will maintain my current score of borderline leaning towards acceptance. Generally, I find the paper to be sound in its methodology (and I appreciate the authors' direct and principled responses), but the level of contribution is limited by the incompleteness of the characterization and the narrowness of scope.

---

### Official Review · Reviewer_Cayz · 2025-11-11

**Soundness:** 4
**Presentation:** 3
**Contribution:** 2
**Rating:** 4
**Confidence:** 3

**Summary:**

This paper examines how transformers solve algebraic problems. Unlike the standard setting, however, the meaning of the symbols are not fixed, and must be inferred from their context. In this setting, and unlike previous mech interp work that showed that models used geometric embeddings (famously the Fourier basis) to do arithmetic, models use symbolic reasoning in this setting. The authors use e.g. causal ablations to show that models develop three mechanisms to do this: attention heads to verbatim/commutative copying, a mechanism for identity element recognition, and something they term “closure-based cancellation” where models track groups membership to constrain valid answers.

**Strengths:**

- There are a large number of experimental results. The models design a novel in-context algebra task, and perform lots of experiments. Unlike previous settings, this task isolates in-context arithmetic. It elicits symbolic mechanisms; the authors very convincingly show this.
- Likewise, the authors bring a number of methods to bear to prove out the mechanisms described above, including computing the indirect effects at the level of attention heads.

**Weaknesses:**

- This task is novel, however it strikes me as contrived and very toy. I struggle to see (a) how these findings will generalize to more interesting settings, (b) what this tells us about models that we did not already know, or (c) how this work convincingly tests interpretability methods.

**Questions:**

- How generalizable are the symbolic mechanisms might the symbolic mechanisms identified here? For example, how might these interact with the geometric mechanisms found robustly in prior work? In a mixed task with both fixed-symbol and variable-symbol arithmetic, would the model would the geometric strategy dominate?
- It is maybe an over-claim that the PCA direction "causally controls identity recognition”? Namely, the described modification changes the "query-promotion" component and must be paired with a second, separate intervention (inserting a false identity fact) for the full identity-demotion step.

---

> ### Author Response · Authors · 2025-11-21
>
> Thank you for your review, we’re glad you found the experiments regarding the evidence of symbolic mechanisms convincing. We have updated our paper and address your specific comments and concerns below:
>
> > “The task is novel, but strikes me as very toy. How will these findings generalize to more interesting settings? What does this tell us about models that we did not already know?”
>
> - Our in-context algebra setting is inspired by the fact that in natural language, much of the meaning of every word is determined by the context. However, this is entangled with the prior meaning of words that is learned from training and baked into the token embeddings of LMs. Because of this, it can be difficult to discern the impact of parametric knowledge in LMs and distinguish it from the effect of context. We designed the in-context algebra setting to cleanly isolate in-context reasoning from pre-encoded embedding information.
> - In our setting, removing the ability for tokens to have fixed value meanings showed us that transformer models can reason by manipulating references in-context, without knowing their underlying meaning. This is similar to how high-school algebra students learn to solve math problems by manipulating letter variables rather than constantly thinking about the values they might represent. This suggests it is possible for LMs to use similar symbolic strategies to solve other problems. Previous studies of the choice between “symbol-based” and “value-based” reasoning in LMs have yielded inconsistent results [Calais et al, Cheng et al], though we have seen studies of LMs consistently learning another form of referential reasoning mechanisms (e.g., binding/ordering IDs [Dai et al, Feng et al, Prakash et al.]). The mechanisms we characterize in our work seem to be task-specific, suggesting that there are potentially more referential reasoning mechanisms LMs develop that we are not yet aware of.
>
> > “How does this work convincingly test interpretability models?”
>
> - The goal of our work is not to test interpretability methods per se, but rather to use interpretability methods to discover and understand interesting behaviors that models develop when trained on the in-context algebra setting. In case we misunderstood your question can you please elaborate more about what you mean here by “testing interpretability methods”?
>
> > “How generalizable are the symbolic mechanisms identified here? For example, how might geometric mechanisms interact with symbolic strategies in a mixed task with both fixed-symbol and variable-symbol arithmetic”
>
> - The mechanisms we identify here are largely task-specific, but seem to be algorithmically general and consistently learned across many seeds and training configurations of our task setup (see Appendix B). Regardless of the domain, if the input has a similar form to the data used to initially discover a mechanism, it’s possible to expect a similar symbolic mechanism at play. For example, copying is a symbolic mechanism whose valid “domain”  is any sequence with at least one repetition in-context. While copying has been heavily studied [Olsson et al], it would be interesting to characterize the “domains of invariance” of other symbolic mechanisms.
> - We think the mix between fixed-symbol and variable-symbols you suggested would be an interesting setting to study for future work! We also thought about this as a potential direction to study while working on this paper but ultimately considered it out of scope, as we discovered that understanding reasoning with pure variables was a rich enough setting on its own.
>
> > “Over-claim that PCA ‘causally controls identity recognition’”
> - Thanks for pointing this out. We agree, and have updated the text in Section 5.3 with a more precise statement about the role of the PCA direction. It now says, “Our experiments suggest that the dominant PCA direction in representation space controls the query promotion submechanism.”, and “the learned PCA direction has causal influence over the model's identity reasoning, enabling us to both enhance and suppress identity predictions.”
>
> ___
> For links to citations, see response to all reviewers.

---

### Author Response · Authors · 2025-11-21
**Response to all Reviewers (1/2)**

We would like to thank each of the reviewers for their thoughtful and constructive feedback. We have made several improvements and additions to the manuscript based on reviewer comments. We provide a summary of the key changes we have made here, and also respond to individual reviewers in separate comments.

We are also running more experiments related to questions asked by XJFb, 9LQ5 and will update this thread with more information as they finish.

**Changes to the Main Paper:**

- **Relating to fourier arithmetic representation work:** Our findings are distinct and complementary to prior work studying fourier representations, as we investigate a new setting where tokens lack fixed meaning;  We have updated the Conclusion section to clarify this focus  (Reviewer 9LQ5)

- **Associative Law Modeling (Figure 3):** We have updated Figure 3 in the main paper and corresponding text in Section 4.1 to include associativity coverage (shown in blue) for both train and hold-out data. (Reviewer XJFb, 9LQ5)

- **Mechanism-specific loss curves (now numbered Figure 6):** One of the most interesting findings of the paper is the fact that drops in loss correspond to learning different mechanisms/skills throughout the course of training. We have clarified this by updating Figure 6 and moving it to Section 6 in the main paper, which we have renamed “Phase Transitions Correspond to Learning of Discrete Skills”.

- **Related Work:** Thank you for pointing out additional related work; we have added additional references to both section 7 (Related Work) and the appendix to address reviewer comments (Reviewer Cayz, XJFb, 9LQ5)

- **Correction to Figure 2c:** We wish to clarify for the reviewers that hold-out performance on semigroups is very good, similar to hold-out performance on unseen groups, while hold-out performance on (non-group) quasigroups and magmas is low, suggesting that the model has learned to exploit particular abstract algebraic structure seen in training.  We have updated Figure 2c and corresponding text in Section 3 to show model performance on the three groups of order 8 that were not seen during training. Plots for non-group algebraic structures have been moved to Appendix B.3. (Reviewer 9LQ5)

---

> ### Author Response · Authors · 2025-11-21
> **Response to all Reviewers (2/2)**
>
> **Changes to the Appendix:**
> - 16 additional models sweeping over counts of layers, heads, dimensions, and training data distribution are trained and analyzed; Appendix B includes complete details as well as additional points of comparison on various groups in and out of distribution. (Reviewer sQQh, 9LQ5)
> - The closure-based cancellation mechanism is described in more detail in Appendix D. In particular, we examine how the cancellation set is computed and provide evidence suggesting it is built from the contribution of several attention heads working together. (Reviewer 9LQ5)
> ___
> **How does our in-context algebra task generalize to more interesting settings? (Reviewer Cayz, sQQh, 9LQ5)**
> - Our in-context algebra setting is inspired by the fact that in natural language, much of the meaning of every word is determined by the context. However, this is entangled with the prior meaning of words that is learned from training and baked into the token embeddings of LMs. Because of this, it can be difficult to discern the impact of parametric knowledge in LMs and distinguish it from the effect of context. We designed the in-context algebra setting to cleanly isolate in-context reasoning from pre-encoded embedding information.
> - In our setting, removing the ability for tokens to have fixed value meanings showed us that transformer models can reason by manipulating references in-context, without knowing their underlying meaning. This is similar to how high-school algebra students learn to solve math problems by manipulating letter variables rather than constantly thinking about the values they might represent. This suggests it is possible for LMs to use similar symbolic strategies to solve other problems. Previous studies of the choice between “symbol-based” and “value-based” reasoning in LMs have yielded inconsistent results [Calais et al, Cheng et al], though we have seen studies of LMs consistently learning another form of referential reasoning mechanisms (e.g., binding/ordering IDs [Dai et al, Feng et al, Prakash et al.]). The mechanisms we characterize in our work seem to be task-specific, suggesting that there are potentially more referential reasoning mechanisms LMs develop that we are not yet aware of.
> - The mechanisms we identify here are largely task-specific, but seem to be algorithmically general. Regardless of the domain, if the input has a similar form to the data used to initially discover a mechanism, it’s possible to expect a similar symbolic mechanism at play. For example, copying is a symbolic mechanism whose valid “domain”  is any sequence with at least one repetition in-context. While copying has been heavily studied [Olsson et al], it would be interesting to characterize the “domain of invariance” of other symbolic mechanisms.
>
> ___
> Dai et al. [Representational Analysis of Binding](https://aclanthology.org/2024.emnlp-main.967/)
>
> Feng et al. [How Do Language Models Bind Entities In Context](https://openreview.net/forum?id=zb3b6oKO77)
>
> Prakash et al. [Fine-Tuning Enhances Existing Mechanisms: A Case Study on Entity Tracking](https://openreview.net/forum?id=8sKcAWOf2D)
>
> Cheng et al. [Can LLMs Reason Abstractly Over Math Word Problems Without CoT? Disentangling Abstract Formulation From Arithmetic Computation](https://aclanthology.org/2025.emnlp-main.723/)
>
> Calais et al. [Disentangling Text and Math in Word Problems: Evidence for the Bidimensional Structure of Large Language Models’ Reasoning](https://aclanthology.org/2025.findings-acl.656)
>
> Olsson et al. [In-context learning and induction heads.](https://transformer-circuits.pub/2022/in-context-learning-and-induction-heads/index.html)

---

### Meta-Review · Area_Chair_RBVQ · 2026-01-05

**Summary:**

Reviewers find the paper technically sound and carefully executed, but raise consistent concerns about the limited scope and contribution. The task is seen as narrow and toy-like, with largely simple and task-specific mechanisms, and a key limitation is the lack of mechanistic understanding of associative composition. After the clarifications and added experiments in the rebuttal, concerns are generally addressed. Therefore, this paper could be accepted by ICLR.

**Reviewer Concerns:**

The concerns are generally addressed by the rebuttal.

**Reviewer Scores:**

Reviewer 9LQ5 tends to raise his/her rating to be more positive. Others may maintain their positive ratings.

---

### Decision · Program_Chairs · 2026-01-26

Accept (Poster)